# Dipeptidylpeptidase 4 inhibition attenuates gestational pathologies via immune homeostasis restoration in the pulmonary-uterine axis

Guirong Shi, Shengdi Xi, Mengyuan Lv, Yihang Chen, Yonggang Zhou, Haiming Wei ⓘ ✉ & Binqing Fu ⓘ ✉

Respiratory viral infections during pregnancy threaten maternal pulmonary health and fetal development, however the mechanisms linking lung infection to distant uterine immune disruption remain unclear. Here, we demonstrate that respiratory viral infection attenuates uterine immune activation, impairing vascular remodeling and trophoblast invasion, which compromises embryonic growth. Treatment with the dipeptidylpeptidase 4 (DPP4) inhibitor sitagliptin restores immune homeostasis in both the lung and uterus, markedly reducing pregnancy complications. Mechanistically, while pulmonary infection expands interleukin-1 receptor type 2–expressing (IL1R2[+]) CD11b[+] myeloid cells in the lung, this expansion is attenuated by DPP4 inhibition. These cells migrate to the decidua and disrupt pregnancy-maintaining immune signaling. Single-cell RNA sequencing confirms accumulation of IL1R2[+] regulatory macrophages at both sites. Genetic *Il1r2* ablation similarly reduces uterine IL1R2[+] cells and restores gestation. This study reveals a lung-uterus immune axis and identifies DPP4 inhibition as a dual-organ therapeutic strategy against viral-induced pregnancy pathology.

Respiratory viral infections during pregnancy are associated not only with maternal pulmonary complications but also with severe pregnancy-related outcomes, including intrauterine growth restriction, preterm birth, and neonatal mortality[1–3]. Moreover, such infections are linked to long-term health conditions in offspring, including cardiovascular disease and neurodevelopmental disorders such as schizophrenia[4]. In addition to seasonal and pandemic influenza strains[5], other respiratory pathogens—such as severe acute respiratory syndrome (SARS) coronavirus[6], Middle East Respiratory Syndrome (MERS) coronavirus[7], novel coronavirus (SARS-CoV-2)[8], and respiratory syncytial virus (RSV)[9]—have also been implicated in gestational complications. These associations underscore an urgent clinical need

for effective interventions to mitigate pregnancy abnormalities secondary to maternal respiratory viral infections.

Current research on respiratory viral infections during pregnancy has primarily focused on hormonal fluctuations and viral transmission dynamics[10,11]. However, growing evidence suggests that these infections also disrupt immune function at distal mucosal sites. For instance, influenza A virus and RSV can induce significant alterations in gut microbiota, thereby impairing mucosal immunity[12]. Additionally, they can interfere with intestinal metabolic outputs, modulating the biosynthesis of ligands for innate-like T cells, such as natural killer T (NKT) cells[13], and mucosa-associated invariant T (MAIT) cells[14]. Lung-derived type I interferons produced during influenza infection have

State Key Laboratory of Immune Response and Immunotherapy, Department of Obstetrics and Gynecology, The First Affiliated Hospital of USTC, Center for Advanced Interdisciplinary Science & Biomedicine of IHM, Division of Life Sciences and Medicine, University of Science and Technology of China, Hefei, Anhui, China. ✉e-mail: ustcwhm@ustc.edu.cn; fbq@ustc.edu.cn

also been shown to suppress intestinal immune responses against bacterial pathogens such as *Salmonella*, even in the absence of direct intestinal infection[15]. Despite the uterus being a key mucosal immune organ during pregnancy, whether it is affected by respiratory viral infections has not been elucidated.

The maternal-fetal interface is densely populated by immune cells, primarily of the innate lineage. Decidual natural killer (NK) cells constitute approximately 70% of the immune cell population, followed by macrophages at around 20%[16,17]. Contrary to the long-standing notion that sustained immunosuppression is essential for a successful pregnancy, recent findings highlight the importance of temporally regulated immune activation[18,19]. Embryo implantation requires a proinflammatory environment, characterized by the upregulation of cytokines such as interleukin-6 (IL-6), interleukin-1 (IL-1), and leukemia inhibitory factor (LIF), which are evolutionarily conserved among placental mammals[20,21]. Post-implantation, trophoblast cells secrete chemokines to recruit NK cells and monocytes to the maternal-fetal interface, facilitating extracellular matrix remodeling and uterine spiral artery transformation[22,23]. This finely tuned immune–trophoblast crosstalk, mediated through receptor–ligand interactions, ensures successful trophoblast invasion, vascular remodeling, fetal development, and maintenance of decidual integrity[24,25]. Whether respiratory viral infections disrupt this immune equilibrium in the uterus remains a critical and unexplored question.

In this study, we investigate how respiratory viral infections disrupt uterine immunity and contribute to gestational pathologies. We demonstrate that the DPP4 inhibitor sitagliptin restores fetal development by modulating lung-derived IL1R2[+] myeloid cell migration and decidual immune function. These findings reveal a previously unrecognized lung-uterus immune axis and establish DPP4 inhibition as a potential dual-organ therapeutic strategy for virus-associated pregnancy complications.

## Results

### Respiratory influenza virus infection suppresses immune activation at the maternal-fetal interface

To evaluate immune alterations at the maternal-fetal interface following respiratory influenza virus infection, we established an H1N1 influenza A infection model in pregnant mice at embryonic day 6.5 (E6.5) and monitored pregnancy progression (Supplementary Fig. 1a). Consistent with previous reports[26], infected dams exhibited reduced gestational weight gain (Supplementary Fig. 1b). Macroscopic examination of the lungs revealed dark red lesions and tissue swelling (Supplementary Fig. 1c), and histological analyses showed exacerbated inflammation, fibrosis, and heightened immune cell infiltration (Supplementary Fig. 1d–f). Influenza virus infection is known to damage alveolar epithelial cells, trigger robust immune responses, and induce proinflammatory cytokine release[27]. Accordingly, expression levels of *Il6*, *Tnf*, and *Il1b* were significantly elevated in lung tissues following infection (Supplementary Fig. 1g). Fetal assessments at E12.5 and E16.5 revealed marked intrauterine growth restriction, evidenced by significantly reduced fetal body length and weight in infected pregnancies (Supplementary Fig. 1h–k). These findings confirm that respiratory influenza infection during early gestation causes severe pulmonary pathology and fetal growth impairment, validating the robustness of our infection model.

Respiratory viruses rarely traverse the placental barrier[28]. Consistent with this, viral RNA was detected exclusively in lung tissues, with no detectable viral gene expression in decidual or placental tissues (Supplementary Fig. 1l), aligning with previous findings. Additionally, type I interferons were significantly upregulated in the lungs but not in the decidua or placenta (Supplementary Fig. 1m, n), indicating that fetal growth restriction is not due to direct viral infection at the maternal-fetal interface.

Transcriptomic analysis of decidual tissues revealed significant alterations in immune signaling pathways. Immune-inhibitory pathways—including negative regulation of innate immune responses and cytokine production—were upregulated, while immune-activating pathways were suppressed post-infection (Fig. 1a, b). Notably, several cytokines upregulated in the lung—IL-28, MIP-3α (Ccl20), IL-6, and IL-21—were downregulated in the decidua following infection (Fig. 1c, d). We also observed significant upregulation of the type I interferon signaling inhibitor *Usp18* in decidual tissues (Fig. 1e). Usp18 attenuates immune responses by deconjugating ISG15 from its substrates, thereby suppressing inflammatory cytokine production[29]. These findings suggest that respiratory influenza infection induces a systemic immunosuppressive state at the maternal-fetal interface.

Successful pregnancy requires localized immune activation to enable extracellular matrix remodeling and trophoblast invasion—key processes in spiral artery remodeling. VEGF and MMP9 are essential mediators of angiogenesis and tissue remodeling[30]. Following infection, both *Vegf* and *Mmp9* expression were significantly reduced in decidual tissues (Fig. 1f, g), indicating compromised angiogenesis. Histological analysis further revealed impaired vascular remodeling, with increased vascular wall thickness and decreased cavity area in infected deciduas (Fig. 1h, i). Cytokeratin staining showed a reduction in trophoblast cell invasion (Fig. 1j), a critical process for adequate spiral artery remodeling and placental perfusion[30]. These data collectively demonstrate that respiratory influenza infection leads to suppressed immune activation at the maternal-fetal interface, impairs vascular remodeling and trophoblast invasion, and ultimately restricts fetal growth.

### DPP4 inhibition rescues pregnancy outcomes following respiratory virus infection

We next investigated the mechanisms linking pulmonary infection to uterine immune dysregulation and evaluated potential therapeutic strategies. Given that the respiratory and reproductive systems are interconnected through systemic circulation, we hypothesized that blood-borne factors may mediate communication between the lungs and uterus. Proteomic analysis of serum collected at 12.5 days of gestation (6 days post-infection) revealed a significant increase in DPP4 protein levels, which was validated by ELISA (Fig. 2a, b). DPP4, a dipeptidyl peptidase expressed in both membrane-bound and soluble forms, is implicated in immune modulation, inflammation, and metabolic regulation[31,32].

To assess therapeutic potential, we administered the DPP4 inhibitor sitagliptin to infected dams by daily oral gavage from E7.5 to E11.5, while controls received vehicle (Fig. 2c). Sitagliptin treatment significantly mitigated virus-induced maternal weight loss (Fig. 2d) and reduced pulmonary injury. Macroscopic examination of the lungs revealed dark red lesions and tissue swelling (Fig. 2e) and histopathological analyses demonstrated reduced pulmonary inflammation in the H1N1+Sitagliptin group (Fig. 2f, g).

Treated embryos displayed increased weight and body length, along with enlarged placental diameter and weight compared to untreated infected controls (Fig. 2h, i). Histological analysis showed that sitagliptin restored angiogenesis at the maternal-fetal interface. Vascular wall thickness and cavity area returned to levels comparable to uninfected controls, and trophoblast invasion was reinstated (Fig. 2j–l). Expression of *Vegfa* also increased following treatment, approximating the levels observed in uninfected pregnancies (Fig. 2m). Taken together, these results demonstrate that DPP4 inhibition via sitagliptin ameliorates the adverse effects of respiratory influenza virus infection on pregnancy by restoring immune and vascular homeostasis at the maternal-fetal interface.

To investigate whether the protective effect of sitagliptin was mediated through immunomodulation rather than via suppression of viral replication, we performed immunohistochemical staining for influenza nucleoprotein, viral titer assays, and quantification of viral gene expression in both lung and decidual tissues following influenza

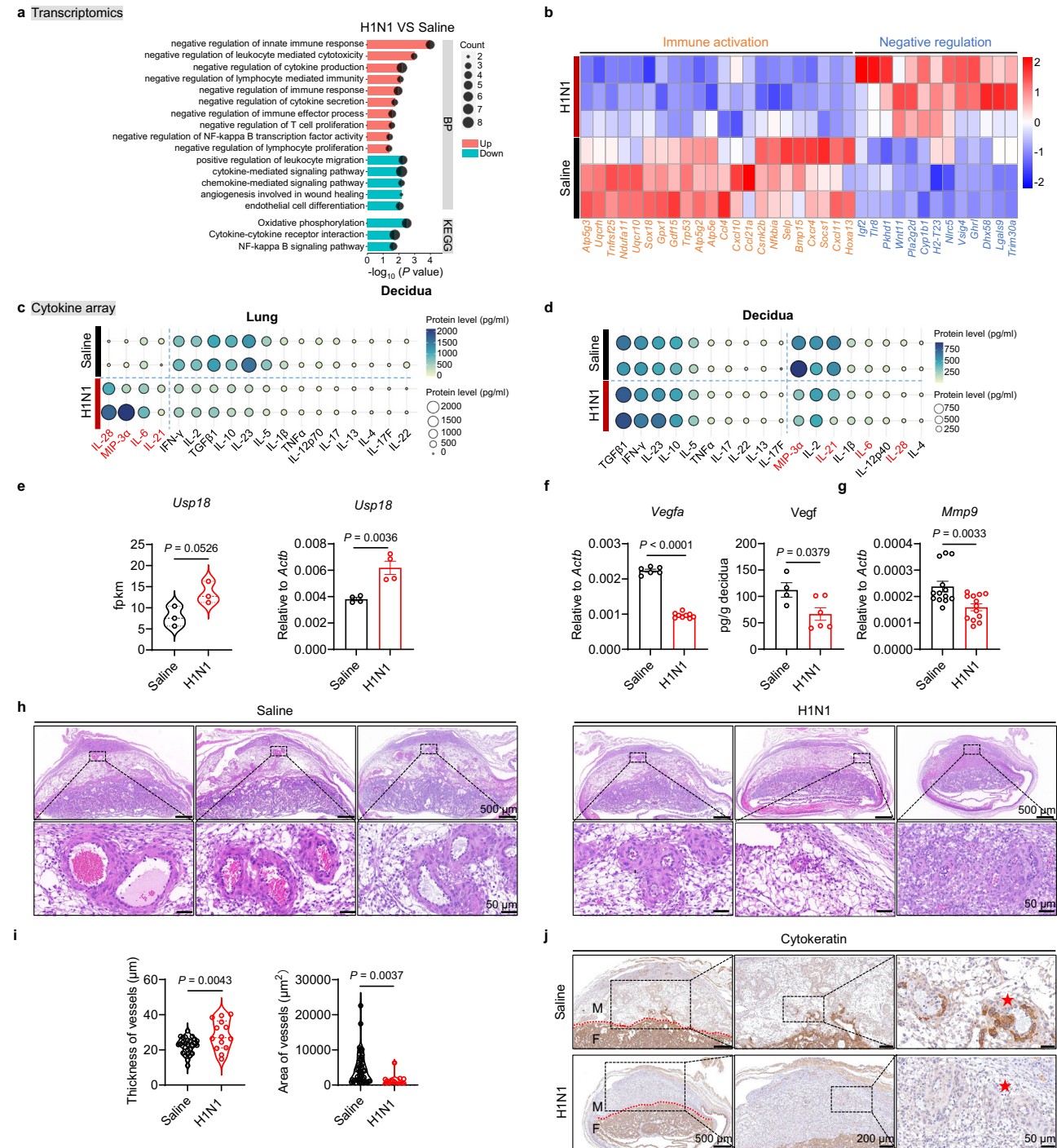

**Fig. 1 | Respiratory influenza virus infection results in attenuated immune activation in the decidua. a–j** Maternal lungs and deciduas were collected at embryonic day 12.5 (E12.5) for analysis. **a** Gene Ontology (GO) of Biological Process (BP) and Kyoto Encyclopedia of Genes and Genomes (KEGG) pathway enrichment analysis of deciduas infected with either saline ($n = 3$) or H1N1 ($n = 3$). **b** Gene expression heatmap for immune activation and negative regulation in deciduas infected with either saline ($n = 3$) or H1N1 ($n = 3$). **c**, **d** Cytokine profiles in lungs (**c**) and deciduas (**d**) infected with either saline or H1N1 assessed with a Mouse Cytokine Antibody Array. The red font denotes four cytokines showing opposite expression between lungs and deciduas. **e** Fpkm values and Quantitative PCR (qPCR) analysis of *Usp18* expression in deciduas infected with either saline ($n = 4$) or H1N1 ($n = 4$). **f** qPCR and enzyme-linked immunosorbent assay (ELISA) analysis of Vegf expression in deciduas infected with either saline ($n = 6$ for qPCR and $n = 4$ for ELISA) or H1N1 ($n = 9$ for qPCR and $n = 6$ for ELISA). **g** qPCR analysis of *Mmp9* expression in

deciduas infected with either saline ($n = 13$) or H1N1 ($n = 13$). **h**, **i** Representative images of H&E staining of vessels in decidua sections (**h**) and statistical analysis of vessel thickness and area (**i**) infected with either saline ($n = 27$ for thickness and $n = 47$ for area) or H1N1 ($n = 14$ for thickness and $n = 16$ for area); scale bar, 50 μm. **j** Representative images of immunohistochemistry staining of Cytokeratin in decidua sections infected with either saline or H1N1. scale bar, 50 μm. The red line indicates the boundary between the maternal and fetal sides. The red star marks the site of trophoblast invasion into the decidua. M, maternal; F, fetal. Results are representative of two or three independent experiments. All bars in the graphs represent the mean ± s.e.m. Statistical significance of pathway enrichment (**a**) was determined by the hypergeometric test (one-sided), with *P* values adjusted for multiple comparisons using the false discovery rate (FDR) method. Statistical comparisons were performed using a two-tailed unpaired Student's *t* test (**e**, **f**, **g**, **i**). Source data are provided as a Source Data file.

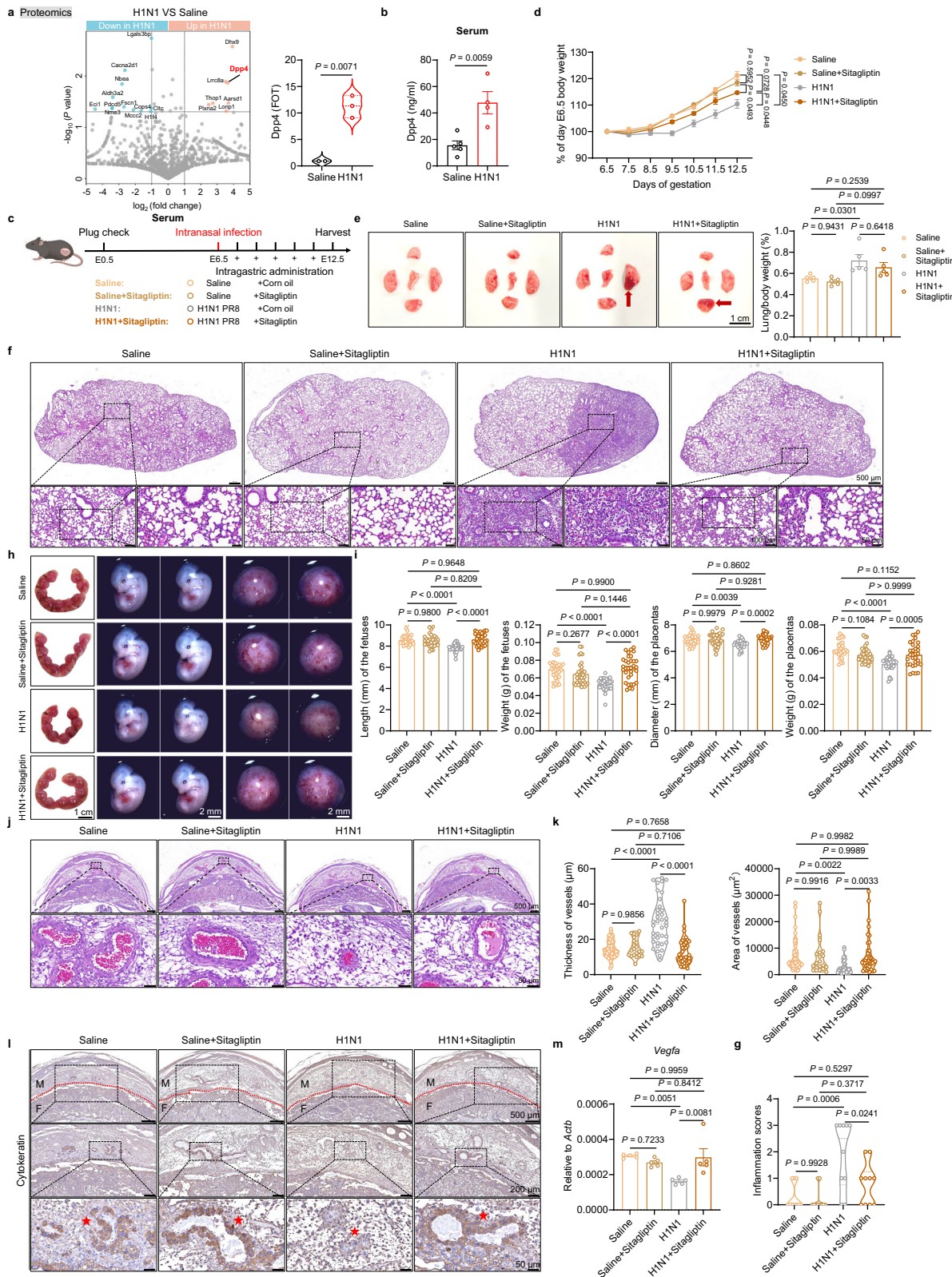

virus infection. These analyses consistently demonstrated that DPP4 inhibitor did not significantly affect viral replication or propagation (Supplementary Fig. 2a–e). Given the role of IL-6 as a key marker of virus-induced lung inflammation[33], we assessed *Il6* expression and found it significantly reduced with sitagliptin treatment compared to the H1N1 group (Supplementary Fig. 2d). These results suggest that while sitagliptin tempers hyperactive immune responses in the lung

and restores immune homeostasis in the decidua, it does not impair antiviral defense. Importantly, separate administration of the inhibitor to healthy pregnant mice revealed no detectable adverse impacts on maternal well-being or prenatal development (Fig. 2d–m and Supplementary Fig. 2a–e), thus establishing its safety profile. These findings confirm its potential as a safe therapeutic candidate for respiratory virus infection in pregnancy. Furthermore, compared with uninfected

**Fig. 2 | Inhibition of DPP4 alleviates intrauterine growth restriction caused by respiratory influenza virus infection. a, b** Maternal serum was collected at embryonic day 12.5 (E12.5) for analysis. **a** Proteomic analysis of differentially expressed proteins in serum from pregnant mice infected with either saline ($n = 2$) or H1N1 ($n = 3$). **b** ELISA analysis of Dpp4 expression in serum infected with either saline ($n = 5$) or H1N1 ($n = 4$). **c** Schematic diagram illustrating the timeline of H1N1 infection and sitagliptin treatment in pregnant mice (Created in BioRender. Ding, X. (2026) https://BioRender.com/lfokc6w). **d** Body weight changes in pregnant mice infected with saline alone ($n = 5$) or treated with sitagliptin ($n = 5$), and infected with H1N1 alone ($n = 5$) or treated with sitagliptin ($n = 5$). **e–m** Maternal lungs and deciduas were collected at E12.5 for analysis. **e** Representative images of lungs infected with saline alone ($n = 5$) or treated with sitagliptin ($n = 5$), and infected with H1N1 alone ($n = 5$) or treated with sitagliptin ($n = 5$) (left). The ratio of lung tissue mass to mouse body weight (right). scale bar, 1 cm. The red arrow marks the parenchymal lung lesion. **f, g** Representative images of hematoxylin and eosin (H&E) staining of lung sections infected with saline alone or treated with sitagliptin, and infected with H1N1 alone or treated with sitagliptin (**f**). scale bar, 50 μm. Inflammation score for lung sections (**g**). **h, i** Representative images of the uterus, fetuses, and placentas (**h**) and statistical analysis of the length and weight of fetuses and the diameter and weight of placentas (**i**) infected with either saline alone ($n = 32$ for length of fetuses, $n = 32$ for weight of fetuses, $n = 32$ for diameter of placentas and $n = 32$ for weight of placentas) or treated with sitagliptin ($n = 33$ for length of fetuses, $n = 33$ for weight of fetuses, $n = 33$ for diameter of placentas and $n = 33$ for weight of placentas), and infected with H1N1 ($n = 32$ for length of fetuses, $n = 32$ for weight of fetuses, $n = 32$ for diameter of placentas and $n = 32$ for weight of placentas), or treated with sitagliptin ($n = 33$ for length of fetuses, $n = 33$ for weight of fetuses, $n = 32$ for diameter of placentas and $n = 32$ for weight of placentas). scale bar, 1 cm. **j, k** Representative images of H&E staining of vessels in decidua sections (**j**) and statistical analysis of vessel thickness and area (**k**) infected with saline alone ($n = 59$ for thickness and $n = 59$ for area) or treated with sitagliptin ($n = 21$ for thickness and $n = 20$ for area), and infected with H1N1 ($n = 38$ for thickness and $n = 38$ for area) alone or treated with sitagliptin ($n = 39$ for thickness and $n = 41$ for area). scale bar, 50 μm. **l** Representative images of immunohistochemistry staining of Cytokeratin in decidua sections infected with saline alone or treated with sitagliptin, and infected with H1N1 alone or treated with sitagliptin. scale bar, 50 μm. The red line indicates the boundary between the maternal and fetal sides. The red star marks the site of trophoblast invasion into the decidua. M, maternal; F, fetal. **m** qPCR analysis of *Vegfa* expression in deciduas infected with saline alone ($n = 5$) or treated with sitagliptin ($n = 5$), and infected with H1N1 alone ($n = 5$) or treated with sitagliptin ($n = 5$). All bars in the graphs represent the mean ± s.e.m. Statistical comparisons were performed using a two-tailed unpaired Student's *t* test (**a**, **b**) and one-way ANOVA with Tukey's multiple comparisons test (**d**, **e**, **g**, **i**, **k**, **m**). Source data are provided as a Source Data file.

controls, inhibitor treatment alone did not significantly change lung neutrophil or macrophage levels, indicating that the DPP4 inhibitor itself does not perturb the baseline pulmonary immune landscape. H1N1 infection, as expected, induced a pronounced increase in both neutrophils and macrophages, which was markedly attenuated by sitagliptin treatment (Supplementary Fig. 2f–i). These results reinforce the conclusion that sitagliptin exerts protection primarily by modulating the pulmonary immune environment.

## DPP4 inhibition has a persistent protective effect towards pregnancy outcomes

To evaluate the long-term effects of sitagliptin treatment, pregnant mice were monitored until successful delivery following a 5-day therapeutic regimen. Maternal body weight was measured at two time points: immediately following treatment completion (embryonic day 11.5) and one day before expected delivery (embryonic day 18.5). Our findings demonstrate that sitagliptin treatment significantly alleviated maternal weight loss induced by the infection (Supplementary Fig. 3a). Post-delivery monitoring revealed that infected mothers exhibited lower body weight at postpartum week 1 and 2, a phenotype mitigated by sitagliptin. By the third week postpartum, maternal weight gain became comparable across all groups (Supplementary Fig. 3b). Neither infection nor treatment affected the timing of parturition (Supplementary Fig. 3c). Importantly, influenza virus infection markedly reduced the number of successfully delivered offspring. The litter size from healthy pregnancies was generally above 6 pups, whereas infection without treatment resulted in fewer than 5 surviving pups. Sitagliptin treatment substantially reversed this impairment, restoring litter sizes to levels comparable to those of uninfected dams (Supplementary Fig. 3d). These results further suggest that in untreated infections, some fetuses sustained severe damage during late gestation, preventing their successful birth. Therefore, the surviving pups from infected but untreated mothers did not show significant differences in health status compared to pups from either uninfected or treated mothers (Supplementary Fig. 3e). Collectively, these findings demonstrate that sitagliptin confers sustained benefits in both maternal recovery and offspring development, supporting its long-term protective efficacy.

Additionally, to assess the diversity of effective treatment windows for the inhibitor, mice were intranasally infected with the influenza virus after the establishment of the choriovitelline placenta at embryonic day 12.5 (E12.5)[34]. This was followed by a 5-day treatment regimen with sitagliptin administered daily by oral gavage from E13.5 to E17.5, while control groups received the vehicle only (Supplementary Fig. 4a). Sitagliptin treatment significantly mitigated maternal weight loss induced by the virus (Supplementary Fig. 4b) and reduced pulmonary injury. Macroscopic examination of the lungs showed dark red lesions and tissue swelling in infected animals, which were markedly improved in the sitagliptin-treated group (Supplementary Fig. 4c, d). At the embryonic level, compared with untreated infected controls, dams treated with sitagliptin exhibited increased fetal weight and body length (Supplementary Fig. 4e, f). Collectively, these findings demonstrate that sitagliptin administration after placental establishment effectively mitigates the detrimental effects of influenza infection on both maternal health and embryonic development.

## DPP4 inhibition ameliorates immune dysregulation in the uterus and lung

We next examined whether DPP4 inhibition could alleviate the immune dysregulation at the maternal-fetal interface induced by respiratory viral infection. Decidual tissues were collected at 6 days post-infection (embryonic day 12.5) from Saline, H1N1, and H1N1+Sitagliptin groups for transcriptome profiling. Volcano plot analysis revealed that genes significantly downregulated by H1N1 infection, such as *Lrrn4*, *Upk3b*, *Upk1b*, and *Car3*, were restored in expression in the H1N1+Sitagliptin group (Supplementary Fig. 5a, b). Conversely, genes upregulated by H1N1 infection, including *Prl7a1*, *Ctsm*, and *Gsdma*, were suppressed by sitagliptin treatment (Supplementary Fig. 5a, b).

Gene ontology (GO) enrichment analysis showed that pathways related to type I interferon signaling and immunosuppressive responses, which were activated by H1N1 infection, were downregulated following sitagliptin treatment (Supplementary Fig. 5c, d). Heatmap analysis further demonstrated that the expression profiles of genes associated with interferon-β response, angiogenesis, antigenic stimulation, and extracellular matrix organization in the H1N1+Sitagliptin group resembled those of the uninfected Saline group, in contrast to the H1N1-infected group (Supplementary Fig. 5e). Notably, expression of *Usp18*, a negative regulator of type I interferon signaling, was significantly reduced with sitagliptin treatment compared to H1N1 infection alone (Supplementary Fig. 5f). Collectively, these findings suggest that DPP4 inhibition restores immune homeostasis at the maternal-fetal interface, supporting normal angiogenesis and promoting a favorable pregnancy environment under conditions of maternal respiratory viral infection.

To determine whether DPP4 inhibition also modulates immune responses in the lung, we performed transcriptomic analysis on lung tissues collected at the same time point. KEGG pathway analysis revealed that several immune-activating pathways—such as PI3K-Akt signaling, ECM-receptor interaction, and leukocyte transendothelial migration—were significantly upregulated in response to H1N1 infection but were suppressed by sitagliptin treatment (Supplementary Fig. 5g, h). These data indicate that sitagliptin mitigates aberrant immune activation in the lung.

## DPP4 inhibition reduces IL1R2 accumulation induced by respiratory viral infection

To elucidate the mechanism by which sitagliptin modulates immune responses in both the uterus and lung, we hypothesized the involvement of a secreted factor produced in the inflamed lung that accumulates at the maternal-fetal interface. GO analysis of lung transcriptomes following H1N1 infection revealed significant enrichment of cytokine production and secretion pathways, particularly those involving the IL-1 family (Fig. 3a). The IL-1 family consists of 11 ligands (including seven active agonists), three receptor antagonists, and one anti-inflammatory cytokine, signaling through a network of six receptor chains, four signaling receptor complexes, two decoy receptors, and two negative regulators[35]. Within this pathway, the decoy receptor *Il1r2*, the antagonist *Il1rn*, the active ligand *Il1b*, and the accessory protein *Il1rap* were identified as key components upregulated post-infection (Fig. 3b). Expression analysis confirmed a significant upregulation of *Il1r2*, *Il1b*, and *Il1rn* in the lung, while *Il1rap* was undetectable (Fig. 3c and Supplementary Fig. 1g).

We next investigated whether Il1r2, Il1b, or Il1rn accumulated in the decidua. Proteomic profiling of decidual tissue identified IL1R2 as one of the most significantly upregulated proteins with known roles in both immune modulation and embryonic development. Notably, IL1R2 exists in both membrane-bound and soluble forms[36] (Fig. 3d). Elevated IL1R2 levels in H1N1-infected decidua were further validated by ELISA (Fig. 3e). As the first-described decoy receptor[37], IL1R2 structurally mimics IL-1R1, competitively inhibiting IL-1 signaling. Immunostaining showed that IL1R2-positive cells were distributed across both the decidua basalis (DB) and mesometrial lymphoid aggregate of pregnancy (MLAp) layers[16,38] (Fig. 3f).

We then assessed whether sitagliptin could attenuate IL1R2 accumulation. Lung tissues from all three groups were analyzed, revealing a marked reduction in *Il1r2* expression in the H1N1+Sitagliptin group (Fig. 3g). Immunofluorescence staining further confirmed reduced IL1R2 protein levels in the lungs of sitagliptin-treated animals (Fig. 3h). Together, these findings indicate that IL1R2 is induced in the lung during viral infection and subsequently accumulates at the maternal-fetal interface. DPP4 inhibition suppresses IL1R2 production in the lung, implicating this decoy receptor as a key mediator linking pulmonary inflammation to uterine immune dysregulation during respiratory viral infection.

## IL1R2+ myeloid cells systemically expand in the lung, blood, and uterus following respiratory viral infection

Given the critical role of IL1R2 in mediating adverse pregnancy outcomes during respiratory influenza virus infection, we hypothesized that IL1R2 is not locally induced in the uterus but instead originates in the lung and traffics to the decidua via the circulatory system. IL1R2 is constitutively expressed in myeloid cells, including neutrophils, monocytes, and macrophages[39]. To investigate its systemic dynamics, we collected lung, peripheral blood, and decidual samples and analyzed both immune cell composition and IL1R2 expression by flow cytometry (Supplementary Fig. 6a).

At 4 days post-infection (E10.5), we observed a significant expansion of CD11b+ myeloid cells—including monocytes, neutrophils, and granulocytes—in the peripheral blood (Fig. 4a). These CD11b+ cells displayed markedly elevated IL1R2 expression as early as 2 days post-infection (E8.5), with sustained upregulation at 6 days post-infection (E12.5) (Fig. 4b, c).

To further assess tissue-specific IL1R2 expression, we examined immune cell populations in the lung, peripheral blood and decidua at E12.5. Infection induced a robust expansion of CD11b+ myeloid cells (macrophages, neutrophils and monocytes) in the lung, accompanied by a similar increase percentage in circulating neutrophils and monocytes (Supplementary Fig. 6b–e). By contrary, we did not observe expansion of CD11b+ myeloid cells in the decidua (Supplementary Fig. 6f, g). Moreover, IL1R2 expression on CD11b+ myeloid cells in the lung was robustly upregulated following H1N1 infection, with expression levels significantly exceeding those observed in the decidua. Nonetheless, in the decidua, IL1R2 was also elevated on monocytes, neutrophils and granulocytes (Fig. 4d, e). Co-immunofluorescence for CD11b and IL1R2 confirmed the presence of double-positive cells in both lung and decidual tissues post-infection (Fig. 4f, g). Together, these findings establish that respiratory influenza infection induces a systemic expansion of IL1R2-positive myeloid cells, with significant enrichment at the maternal-fetal interface.

## Single-cell RNA-seq identifies tissue-homing IL1R2+ macrophages that link pulmonary infection to maternal-fetal interface

To trace the origin of IL1R2-expressing cells at the maternal-fetal interface, we performed single-cell RNA sequencing on CD45+ immune cells isolated from the lungs and decidua of pregnant mice at E12.5 following treatment with saline, H1N1 infection alone, or H1N1 infection plus sitagliptin. After quality control and doublet removal, unsupervised clustering of lung immune cells identified nine distinct subpopulations (Supplementary Fig. 7a, b). Macrophages constituted the most abundant subset and were significantly expanded following H1N1 infection, an effect that was substantially reversed by sitagliptin treatment (Supplementary Fig. 7c). Given their tissue residency and specific expression of the macrophage markers *Ms4a6c* (lung) and *Ms4a4c* (decidua), we annotated this *Lyz2*-sharing population as macrophages, while not excluding the possibility of some monocyte presence. Further analysis revealed a marked H1N1-induced upregulation of *Il1r2* in lung macrophages, which was also significantly suppressed by sitagliptin (Supplementary Fig. 7d). In the decidua, we identified ten immune cell clusters (Supplementary Fig. 7e, f). Analysis of the major macrophage population showed a congruent pattern: H1N1 infection robustly induced *Il1r2* expression, and this induction was attenuated by sitagliptin (Supplementary Fig. 7g, h). These findings underscore macrophages as a key *Il1r2*-expressing population during influenza infection in pregnancy.

We next re-clustered macrophages and searched for the most relevant macrophage subpopulations. Based on macrophage types summarized previously[40], lung macrophages were categorized into six distinct subsets (Fig. 5a). Among these, a subpopulation of immunoregulatory macrophages highly expressed *Il1r2* (Fig. 5b, c). This subset was rare under homeostatic conditions but expanded significantly after H1N1 infection, with its proportion subsequently normalized by sitagliptin (Fig. 5d). Visualization of *Il1r2* expression confirmed its specific enrichment in this immunoregulatory subset after infection, which was abrogated by treatment, suggesting these cells may serve as a key target of sitagliptin and represent a primary source of IL1R2 (Fig. 5e, f). An analogous immunoregulatory macrophage subset, also exhibiting high *Il1r2* expression, was identified in the decidua (Fig. 5g–i). Its abundance increased after infection and decreased with sitagliptin treatment, and its *Il1r2* expression mirrored this trend, closely recapitulating the pulmonary response (Fig. 5j–l). This suggests that these cells may serve as a critical link between the lung and maternal-fetal interface, and constitute a local source of IL1R2.

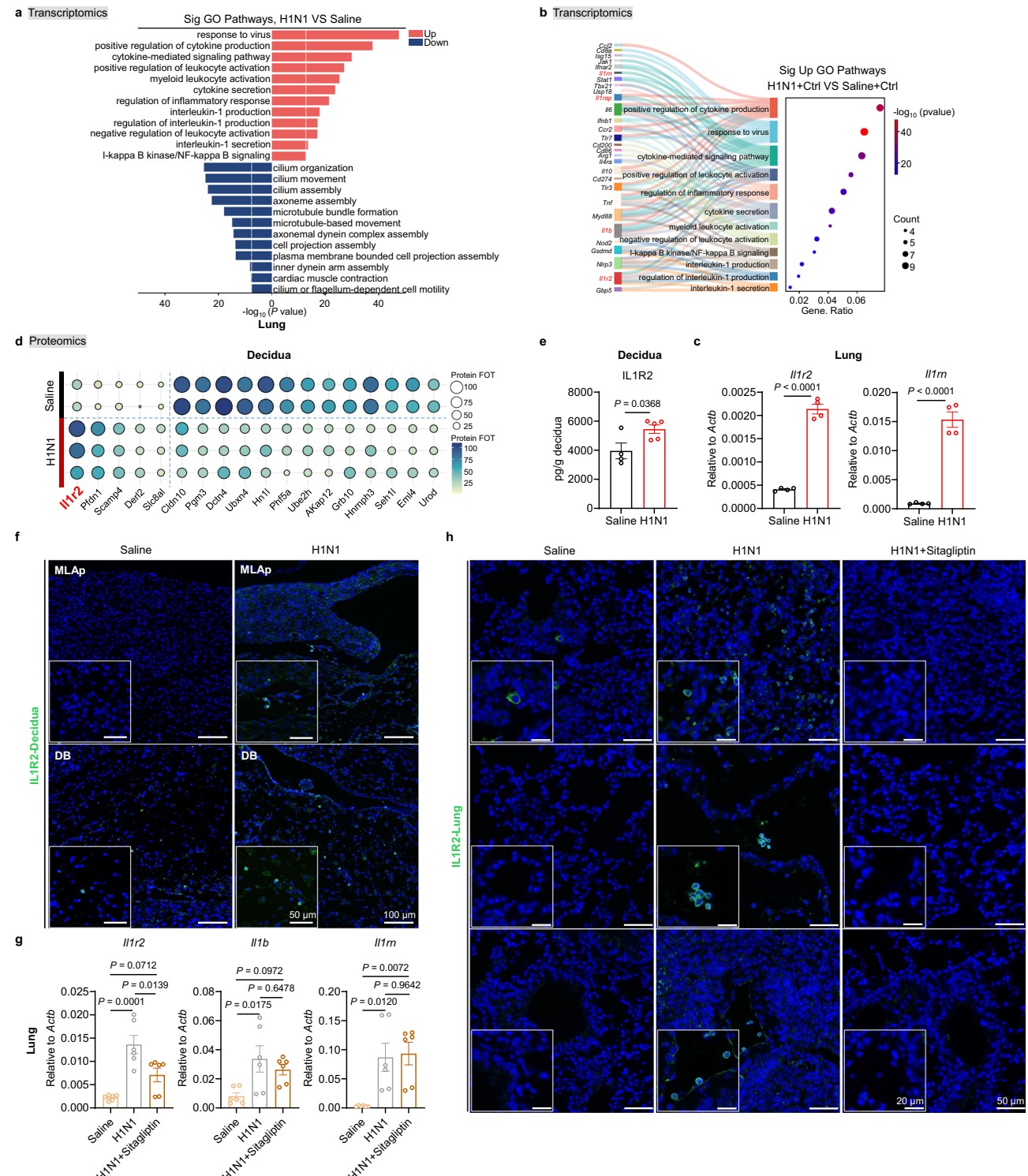

**Fig. 3 | Inhibition of DPP4 reduces the increase of IL1R2 caused by respiratory influenza virus infection. a**–**h** Maternal lungs and deciduas were collected at embryonic day 12.5 (E12.5) for analysis. **a** GO analysis of differentially expressed genes in lungs infected with either saline (*n* = 3) or H1N1 (*n* = 3), or treated with sitagliptin (*n* = 3). **b** Sankey dot plot analysis for pathway enrichment of differentially expressed genes in lungs infected with either saline (*n* = 3) or H1N1 (*n* = 3), or treated with sitagliptin (*n* = 3). **c** qPCR analysis of *Il1r2* and *Il1rn* in lungs infected with either saline (*n* = 4) or H1N1 (*n* = 4). **d** Proteomic analysis of differentially expressed proteins in deciduas infected with either saline (*n* = 2) or H1N1 (*n* = 3). **e** ELISA analysis of IL1R2 expression in deciduas infected with either saline (*n* = 4) or H1N1 (*n* = 5). **f** Representative images of immunofluorescence staining of IL1R2 in decidua sections infected with either saline or H1N1; scale bar, 50 μm. MLAp,

mesometrial lymphocytes associated with pregnancy; DB, decidua basalis. **g** qPCR analysis of *Il1r2*, *Il1b* and *Il1rn* in lungs infected with either saline (*n* = 6) or H1N1 alone (*n* = 6), or treated with sitagliptin (*n* = 6). **h** Representative images of immunofluorescence staining of IL1R2 in lung sections infected with either saline or H1N1 alone, or treated with sitagliptin. scale bar, 20 μm. Results are representative of two or three independent experiments. All bars in the graphs represent the mean ± s.e.m. Statistical significance of pathway enrichment (**a**, **b**) was determined by the hypergeometric test (one-sided), with *P* values adjusted for multiple comparisons using the false discovery rate (FDR) method. Statistical comparisons were performed using a two-tailed unpaired Student's *t* test (**c**, **e**) and one-way ANOVA with Tukey's multiple comparisons test (**g**). Source data are provided as a Source Data file.

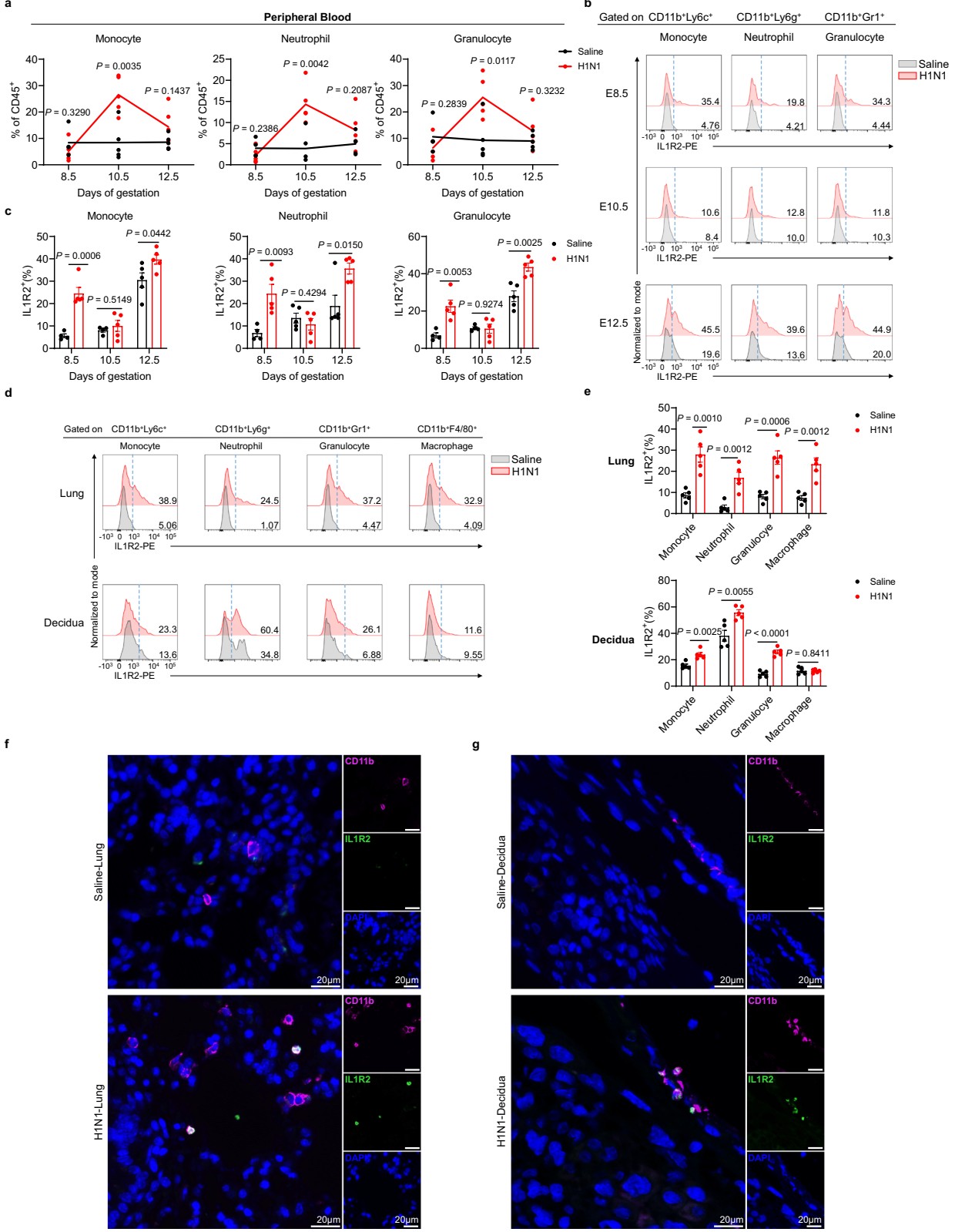

**Fig. 4 | Increased IL1R2 mainly expresses on CD11b⁺ myeloid cells. a** Flow cytometry analysis of the percentages of myeloid cells in peripheral blood infected with either saline (*n* = 4 at E8.5, *n* = 5 at E10.5 and *n* = 5 at E12.5) or H1N1 (*n* = 5 at E8.5, *n* = 5 at E10.5 and *n* = 5 at E12.5) at E8.5, E10.5, and E12.5. **b, c** Representative flow cytometry histograms (**b**) and corresponding quantification of the expression of IL1R2 (**c**) on myeloid cells in peripheral blood infected with either saline or H1N1 (*n* = 4 at E8.5, *n* = 5 at E10.5 and *n* = 5 at E12.5) or H1N1 (*n* = 5 at E8.5, *n* = 5 at E10.5 and *n* = 5 at E12.5) at E8.5, E10.5 and E12.5. **d, e** Representative flow cytometry

histograms (**d**) and corresponding quantification of the expression of IL1R2 (**e**) on myeloid cells in lungs and deciduas infected with either saline (*n* = 5) or H1N1 (*n* = 5) at E12.5. **f, g** Representative images of immunofluorescence staining of CD11b and IL1R2 in lung (**f**) and decidua (**g**) sections infected with either saline or H1N1. scale bar, 20 μm. Results are representative of two or three independent experiments. All bars in the graphs represent the mean ± s.e.m. Statistical comparisons were performed using a two-tailed unpaired Student's *t* test (**a, c, e**). Source data are provided as a Source Data file.

Integrated analysis of genes commonly upregulated in this macrophage subset across tissues revealed a distinct molecular signature: beyond *Il1r2*, these cells specifically expressed notably key mediators of myeloid cell migration and chemotaxis (*Retnlg*, *Cxcr2*, *Mmp9*) and other immunomodulatory genes (*Chil1*, *Steap4*, *Nlrp12*), implying enhanced migratory potential and immunosuppressive ability[41,42] (Fig. 5m, n). In summary, our results indicate that respiratory viral infection induces a specialized subset of immunoregulatory macrophages characterized by high *Il1r2* expression and a migratory gene signature, positioning them as a likely systemic source for the increased IL1R2 observed at the maternal-fetal interface.

### Deletion of *Il1r2* ameliorates virus-induced adverse pregnancy outcomes

Given the significant accumulation of IL1R2 at the maternal-fetal interface following H1N1 infection, we next tested whether IL1R2 mediates virus-induced pregnancy complications. IL1R2 acts as a decoy receptor that sequesters IL-1β, thereby preventing its interaction with IL-1R1 and attenuating downstream signaling[39]. To assess its causal role, we generated *Il1r2*-knockout (KO) mice and mated *Il1r2*-deficient females with wild-type males. Pregnant females were infected with H1N1 at E6.5 and evaluated 6 days later (Fig. 6a). Both *Il1r2*-WT and *Il1r2*-KO mice exhibited severe lung pathology post-infection, characterized by extensive hyperemia and pulmonary enlargement (Fig.6b–e), indicating that *Il1r2* deletion does not impair the host's susceptibility to lung damage.

We then assessed pregnancy outcomes. In WT mice, influenza infection significantly impaired fetal development, as evidenced by reduced embryo length and placental diameter. Strikingly, *Il1r2*-KO mice maintained normal fetal and placental growth post-infection, with no signs of intrauterine growth restriction (Fig. 6f, g). These data suggest that IL1R2 plays a key role in mediating virus-induced fetal growth impairment. Importantly, *Il1r2* deletion under uninfected conditions did not adversely affect embryonic development, underscoring its pathogenic role in the context of infection (Fig. 6f, g).

Histological analysis of the maternal-fetal interface further revealed that *Il1r2* deletion alleviated the vascular remodeling defects induced by infection, as shown by reduced vascular wall thickening (Fig. 6h, i). Additionally, cytokeratin staining demonstrated improved trophoblast invasion in *Il1r2*-KO mice, reversing the impairment seen in infected WT mice (Fig. 6j). We next investigated whether *Il1r2* knockout modulates viral replication. Immunohistochemical staining for influenza nucleoprotein and quantitative analysis of viral gene expression in lung and decidual tissues showed that viral replication and propagation were not significantly affected by *Il1r2* deletion (Supplementary Fig. 8a–c). Collectively, these findings establish IL1R2 as a critical mediator linking respiratory viral infection to adverse pregnancy outcomes and highlight it as a potential therapeutic target for protecting pregnancy during maternal respiratory infections.

### Increase of IL1R2 and fetal growth restriction in the uterus following pulmonary coronavirus infection modeled by intranasal MHV inoculation

To determine whether IL1R2 upregulation is a broader feature of other virus-induced fetal growth restriction, we extended our investigation to a model of coronavirus infection. COVID-19, caused by SARS-CoV-2, presents a major threat to maternal and fetal health during pregnancy[43]. Mouse hepatitis virus (MHV), a member of the same genus as SARS-CoV-2, induces clinical symptoms in mice that closely resemble those observed in COVID-19 patients, including weight loss, multi-organ pathology, and irreversible neurological damage. Therefore, MHV serves as a well-established surrogate model for studying SARS-CoV-2 pathogenesis[44]. Pregnant mice were intranasally infected with MHV at embryonic day 6.5 (E6.5) and monitored through E12.5 (Supplementary Fig. 9a). Infected mice exhibited impaired gestational

weight gain (Supplementary Fig. 9b), as well as significant lung inflammation and enlargement (Supplementary Fig. 9c–e), in line with previous reports[44] and confirming successful model establishment.

Assessment of pregnancy outcomes revealed significant fetal growth restriction following MHV infection, evidenced by reduced embryo weight and length, as well as diminished placental diameter and weight (Supplementary Fig. 9f, g). Histological analysis of the maternal-fetal interface showed impaired angiogenesis, marked by thickened vascular walls (Supplementary Fig. 9h, i). As observed in H1N1 infection (Fig. 1f, g), qPCR analysis revealed reduced expression of key angiogenic factors *Vegfa* and *Mmp9* (Supplementary Fig. 9j), implicating disrupted angiogenesis as a contributor to impaired fetal nutrient transport and subsequent growth restriction.

Interestingly, unlike typical antiviral responses, MHV infection did not elicit a type I interferon response in the lung but instead elevated levels of IFN-γ and pro-inflammatory cytokines such as TNF-α[45]. Viral replication was confined to the lungs, as evidenced by localized upregulation of interferon response and *Tnf*, with no signs of vertical transmission (Supplementary Fig. 9k, l). Notably, expression of inflammatory marker *Tnf* was significantly reduced in the decidua, suggesting attenuated immune activation at the maternal-fetal interface (Supplementary Fig. 9l). These findings parallel the immunological alterations observed in influenza infection and support the notion that nasal inoculation of viruses to simulate pulmonary infection can impair placental function and fetal development via indirect, non-viral mechanisms.

We next examined whether IL1R2 expression is also elevated in this model. Transcriptomic profiling of lung tissue 6 days post-infection revealed significant upregulation of gene pathways related to immune activation, cytokine production, and type II interferon responses—consistent with observed lung pathology. In contrast, pathways involved in ciliary organization and motility were downregulated, indicating disruption of lung epithelial integrity, as seen in influenza models[46] (Supplementary Fig. 10a). Notably, Gene Ontology and Sankey analyses identified upregulation of pathways regulating IL-1 family cytokines, including the immunosuppressive receptor IL1R2, which participates in feedback regulation of inflammatory responses (Supplementary Fig. 10b). FPKM and gene quantification showed markedly increased *Il1r2* expression in lung tissue following infection (Supplementary Fig. 10c). Meanwhile IL1R2 protein levels increased in the decidua (Supplementary Fig. 10d), mirroring our findings in influenza-infected models (Fig. 3c, e). Immunohistochemistry further confirmed the presence of CD11b⁺IL1R2⁺ double-positive cells in both lung and decidual tissues (Supplementary Fig. 10e, f). These results suggest that IL1R2 elevation in the decidua and the associated pregnancy complications may be a common feature across pulmonary infection modeled by intranasal H1N1 or MHV inoculation.

### DPP4 inhibition alleviates pregnancy complications induced by coronavirus infection

To explore potential therapeutic strategies, we investigated whether DPP4 inhibition could mitigate coronavirus-induced pregnancy complications. Transcriptomic data revealed a significant increase in *Dpp4* expression following MHV infection, suggesting a possible pathogenic role and therapeutic target (Fig. 7a, b). Administration of the DPP4 inhibitor sitagliptin significantly attenuated maternal weight loss and reversed fetal growth restriction. Treated embryos exhibited restored body length, weight, and placental size, comparable to uninfected controls (Fig. 7c–e).

To assess whether sitagliptin alleviated immunopathology in maternal tissues, we conducted transcriptomic analyses of lung and decidual tissues post-treatment. In the lung, sitagliptin reversed the downregulation of genes suppressed by infection and significantly decreased expression of pro-inflammatory markers, including *Il1b*, *Nfkbiz*, and *Lyve1* (Supplementary Fig. 10g). GO enrichment analysis

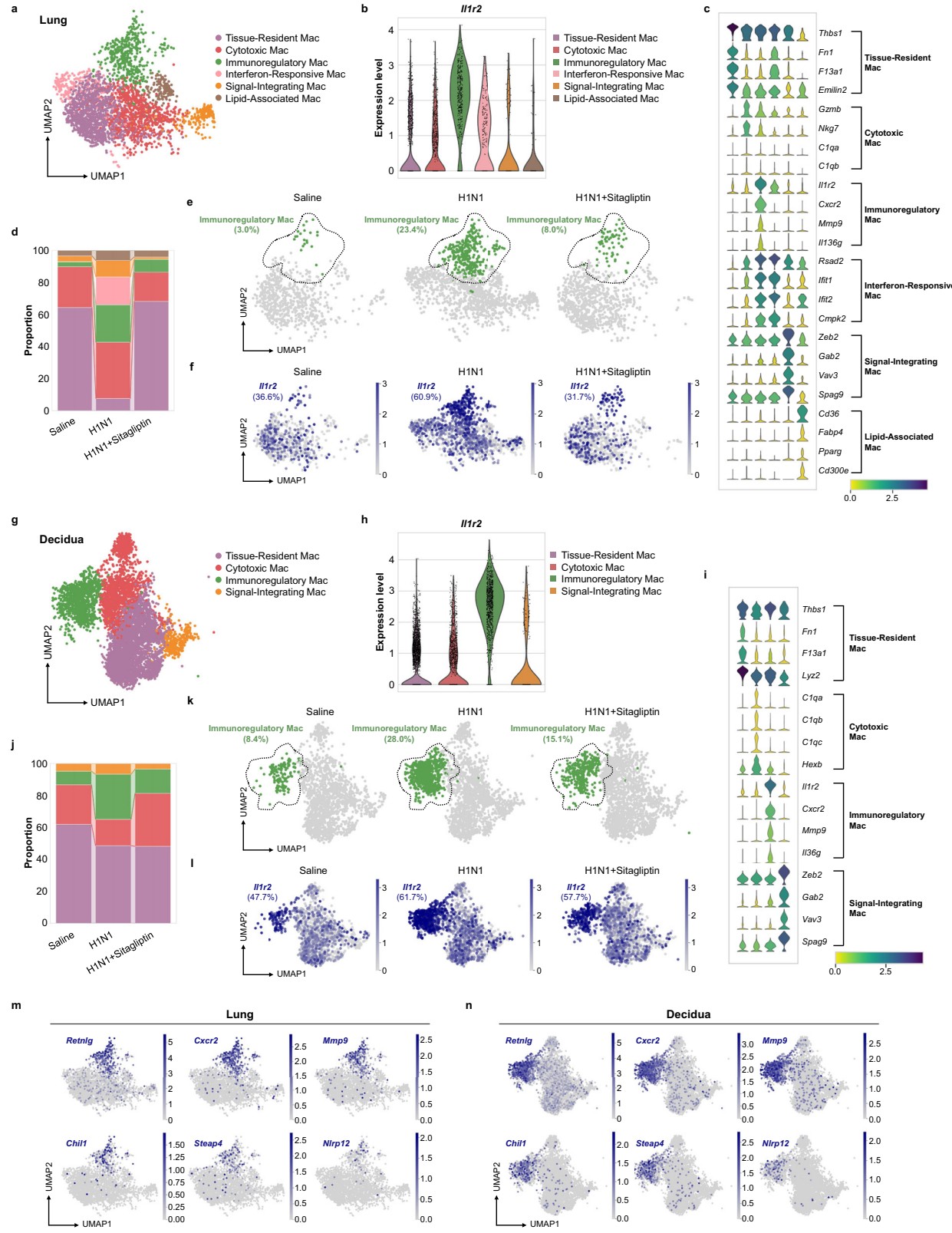

demonstrated recovery of pathways involved in intercellular communication, blood circulation, and tissue homeostasis (Supplementary Fig. 10h), indicating that sitagliptin attenuates excessive lung inflammation.

Histological analysis of the maternal-fetal interface revealed that sitagliptin treatment significantly reduced vascular wall thickening (Fig. 7f, g). Immunohistochemical staining further demonstrated

enhanced trophoblast invasion following treatment (Fig. 7h), indicating a restoration of normal placental development. Notably, sitagliptin also significantly reduced IL1R2 protein levels in the decidua (Fig. 7i), further supporting its therapeutic potential.

To determine whether sitagliptin also restored immune balance at the maternal-fetal interface, we conducted transcriptome sequencing of decidual tissue. Volcano plot analysis revealed that sitagliptin

**Fig. 5 | Inhibition of DPP4 reduces migratory and *Il1r2*-high macrophages following influenza virus infection. a** Subclustering of macrophages in lungs identified six subtypes at 0.3 resolution. A color-coded uniform manifold approximation and projection (UMAP) plot is shown, with each subcluster defined on the right. **b** Violin plots of *Il1r2* expression of each of macrophage cluster in lungs. **c** Violin plots of normalized expression of selected cluster-specific genes in lungs. Four genes are shown per cluster. **d** Proportion of each macrophage cluster in lungs infected with either saline or H1N1 alone, or treated with sitagliptin. **e** Group-separated UMAP plots with Immunoregulatory Mac cluster in green in lungs. **f** UMAP feature plots of *Il1r2* expression of macrophage cluster in lungs infected with either saline or H1N1 alone, or treated with sitagliptin. **g** Subclustering of macrophages in deciduas identified four subtypes at 0.2 resolution. A color-coded uniform manifold approximation and projection (UMAP) plot is shown with each subcluster defined on the right. **h** Violin plots of *Il1r2* expression of each of macrophage cluster in deciduas. **i** Violin plots of normalized expression of selected cluster-specific genes in deciduas. Four genes are shown per cluster. **j** Proportion of each macrophage cluster in deciduas infected with either saline or H1N1 alone, or treated with sitagliptin. **k** Group-separated UMAP plots, with Immunoregulatory Mac cluster in green in deciduas. **l** UMAP feature plots of *Il1r2* expression of macrophage cluster in deciduas infected with either saline or H1N1 alone, or treated with sitagliptin. **m, n** UMAP feature plots of normalized expression of *Retnlg*, *Cxcr2*, *Mmp9*, *Chil1*, *Steap4*, and *Nlrp*12 of macrophage cluster in lungs (**m**) and deciduas (**n**). Source data are provided as a Source Data file.

downregulated multiple genes upregulated by intranasal MHV inoculation such as *Cd40lg*, *Krt8*, and *Nme8*, while upregulating genes suppressed by infection, including *Prl2c5*, *Galnt9*, and *Mmp10* (Fig. 7j). GO analysis indicated that MHV infection triggered immunosuppressive pathways involving cytokines like IL-4, IL-13, and IL-10, while down-regulating key signaling pathways such as JAK-STAT, extracellular matrix organization, and angiogenesis—thereby impairing maternal-fetal communication and pregnancy maintenance. Importantly, sitagliptin treatment reversed many of these effects, enhancing activation of the JAK-STAT pathway and restoring pathways related to immune activity and pregnancy support (Fig. 7k).

To further directly evaluate whether sitagliptin exerts its beneficial effects via antiviral activity in the MHV infection model, we measured viral gene replication in lung tissues after treatment. Critically, sitagliptin did not suppress *MHV* replication in the lungs, and *MHV* remained restricted to the lung without spreading to the maternal-fetal interface (Supplementary Fig. 10i), suggesting that DPP4 may not be involved in MHV amplification and that the receptor for MHV remains CEACAM1, as previously reported[47]. These data clearly demonstrate that the protective effect of sitagliptin on pregnancy outcomes is not mediated by direct antiviral mechanisms, but rather through modulation of the host's immunopathological response.

Collectively, these findings demonstrate that DPP4 inhibition not only mitigates excessive lung inflammation and placental dysfunction caused by coronavirus infection but also counteracts IL1R2 elevation and immune suppression at the maternal-fetal interface. Thus, DPP4 inhibitors such as sitagliptin may represent a promising therapeutic strategy for managing adverse pregnancy outcomes associated with pulmonary infection modeled by intranasal H1N1 or MHV inoculation.

## Discussion

John Bienenstock first proposed the concept of a "common mucosal immune system," wherein mucosal tissues—including the gastro-intestinal, respiratory, and urogenital tracts—are functionally interconnected. This framework suggests that immune activation in one mucosal site can influence distal sites, implying systemic coordination across mucosal surfaces[48]. Yet, a critical question remains unresolved: Can localized immune disturbances in one mucosal tissue directly disrupt immune homeostasis in another?

In this study, we demonstrate that respiratory viral infections during pregnancy compromise immune activation at the maternal-fetal interface, disrupt essential physiological processes such as spiral artery remodeling and trophoblast invasion, and consequently impair embryonic growth. We found that hyperactivation of lung immunity in response to respiratory viruses induces expression of *IL1R2*, a decoy receptor that inhibits IL-1 signaling. IL1R2 protein subsequently accumulates in the uterus, where it disrupts the normal decidual immune response. Therapeutically, both pharmacological suppression of lung inflammation using DPP4 inhibitors and genetic ablation of *Il1r2* reduced uterine IL1R2 protein levels and restored normal pregnancy outcomes. These findings were validated in models of both influenza

and coronavirus infection during pregnancy. Thus, DPP4 emerges as a potential therapeutic target for mitigating adverse pregnancy outcomes linked to respiratory infections. Our findings further illuminate an underexplored axis of communication between respiratory and reproductive mucosal tissues, offering novel insight into the functional landscape of the common mucosal immune system.

Currently, treatment options for influenza in pregnant women are limited. Although timely administration of antiviral agents can offer some protection, their safety profiles during pregnancy remain insufficiently characterized, and circulating drug-resistant strains present an additional challenge[49,50]. DPP4 plays a central role in regulating glucose metabolism and inflammation, acting through both its enzymatic activity and non-enzymatic modulation of immune responses[51]. Gliptins, a class of DPP4 inhibitors, exhibit broad-spectrum anti-inflammatory effects across diverse inflammatory and autoimmune diseases[52,53]. For example, patients with multiple sclerosis, who often display elevated DPP4 expression in T cells, experience reduced disease severity upon DPP4 inhibition. Similar therapeutic benefits have been reported in Crohn's disease and obesity-related inflammation.

In our respiratory virus infection models, transcriptome sequencing of lung tissue revealed pronounced immune hyperactivation, particularly in cytokine production and secretion pathways. Simultaneously, pathways associated with the negative regulation of cell activation were also upregulated. Among IL-1 family molecules, *IL1R2* was markedly induced. Previous studies suggest that *IL1R2* upregulation during acute inflammation may serve as a negative feedback mechanism to limit IL-1β-driven immune overactivation[54]. We observed that lung inflammation and immune activation were attenuated in both H1N1+Sitagliptin and MHV+Sitagliptin groups, supporting the potent anti-inflammatory role of DPP4 inhibition. While DPP4 inhibitors are primarily known for blocking the enzymatic degradation of glucagon-like peptide-1 (GLP-1), their immunomodulatory effects extend beyond GLP-1-dependent mechanisms. These include direct regulation of immune cell function and suppression of proinflammatory cytokine production[55,56]. In the context of respiratory infection, our data suggest that DPP4 inhibitors reduce *IL1R2* expression by dampening excessive pulmonary inflammation. We propose that DPP4 inhibition may reduce IL1R2 accumulation through two non-exclusive mechanisms. First, by preventing the cleavage of chemokines such as CXCL10 and CXCL11, DPP4 inhibition may preserve their canonical functions and prevent the aberrant recruitment of IL1R2⁺ myeloid precursors to inflammatory sites[57]. Second, DPP4 inhibition broadly attenuates the inflammatory milieu, thereby reducing the compensatory drive for IL1R2 upregulation as a feedback brake[58]. Thus, DPP4 inhibition likely acts by resetting the dysregulated immunopathological environment, rather than through a single target, ultimately normalizing this immunosuppressive axis. However, the precise pathways through which DPP4 inhibition modulates *IL1R2* transcription remain to be elucidated.

IL1R2 appears to be the key molecular mediator bridging immune overactivation in the lung and immune suppression in the uterus. Notably, in both influenza and coronavirus infection models, uterine

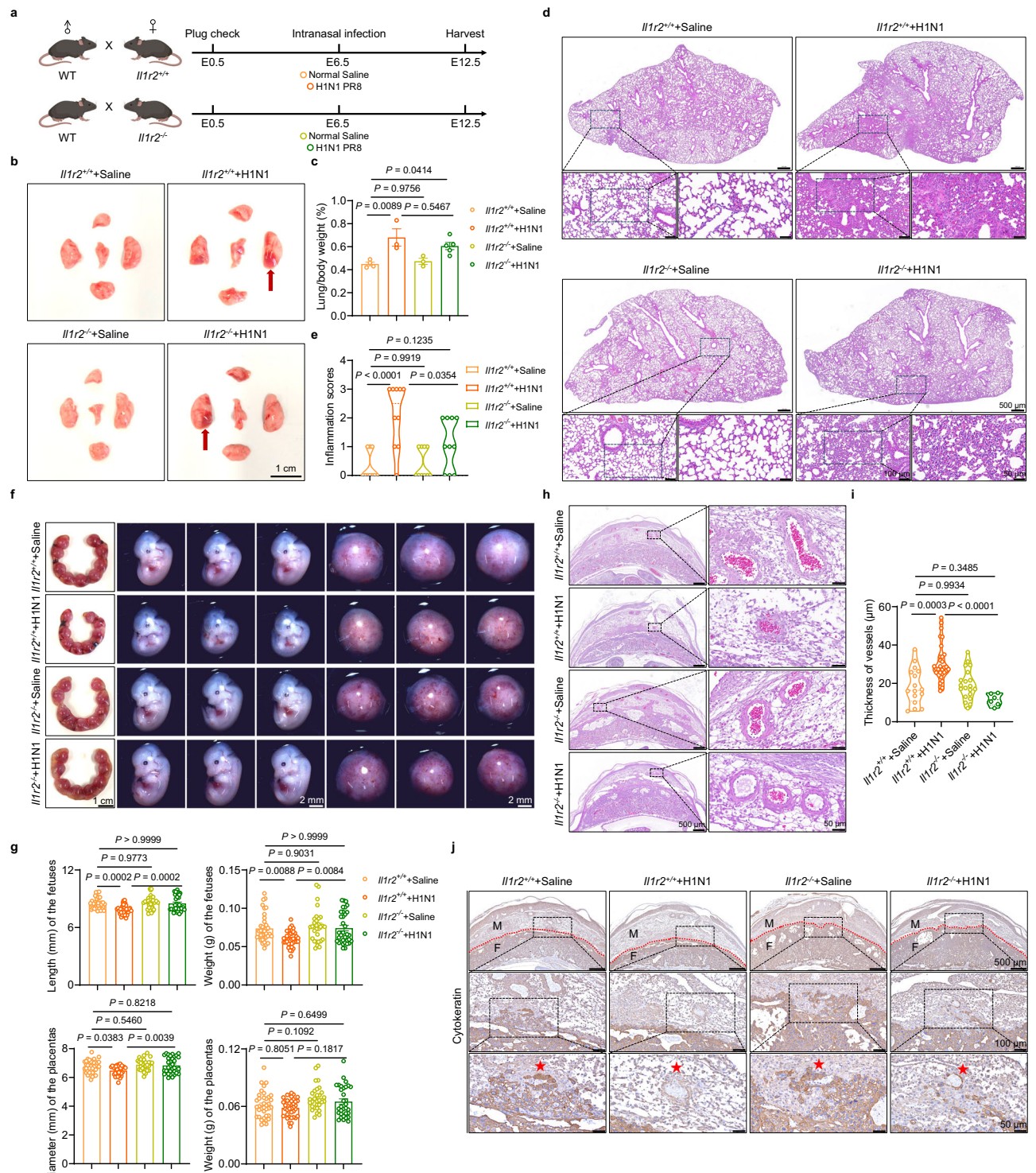

IL1R2 protein levels increased without corresponding detection in virus titers or IFN-β-connected gene expression. Conversely, in lung tissue, viral infection led to elevated *Il1r2* mRNA expression and significantly increased of IL1R2+ myeloid cells, implying that IL1R2 accumulates in the uterus via systemic circulation rather than in situ production. Flow cytometric analyses further confirmed increased IL1R2 expression on myeloid cells within the lung, peripheral blood, and decidua following infection. The gradient—highest in the lung, intermediate in blood, and lowest in the decidua—suggests a migration trajectory of IL1R2-high cells from the lung to the uterus. Our scRNA data further identified tissue-homing IL1R2+ macrophages that link

pulmonary infection to maternal–fetal interface. Together, these findings support a model in which IL1R2 functions as a molecular conduit linking the immune landscapes of the respiratory and reproductive tracts, reinforcing and refining the concept of the common mucosal immune system.

Functionally, IL1R2 acts as a decoy receptor by competitively binding to IL-1 ligands and preventing engagement of IL-1R1, thereby suppressing downstream signaling[59]. IL-1 family cytokines are central to inflammatory and innate immune responses and play key roles in angiogenesis, tissue remodeling, and hematopoiesis[60]. Mechanistically, IL-1 signaling promotes the expression of angiogenic and

**Fig. 6 | Knock out *Il1r2* alleviates intrauterine growth restriction caused by respiratory influenza virus infection. a** Schematic diagram illustrating the mating strategy and timing of infection in pregnant *Il1r2*+/+ and *Il1r2*−/− mice (Created in BioRender. Ding, X. (2026) https://BioRender.com/lfokc6w). **b**–**j** Maternal lungs and deciduas were collected at embryonic day 12.5 (E12.5) for analysis. **b**, **c** Representative images of lungs from pregnant *Il1r2*+/+ (*n* = 4 for Saline and *n* = 3 for H1N1) and *Il1r2*−/− (*n* = 3 for Saline and *n* = 5 for H1N1) mice (**b**) and the ratio of lung tissue mass to mouse body weight (**c**). scale bar, 1 cm. The red arrow marks the parenchymal lung lesion. **d**, **e** Representative images of hematoxylin and eosin (H&E) staining of lung sections (**d**) and inflammation score for lung sections (**e**) from pregnant *Il1r2*+/+ and *Il1r2*−/− mice infected with either saline or H1N1. scale bar, 50 μm. **f**, **g** Representative images of the uterus, fetuses, and placentas (**f**) and statistical analysis of the length and weight of fetuses and the diameter and weight of placentas from pregnant *Il1r2*+/+ (*n* = 36 for length of fetuses, *n* = 36 for weight of fetuses, *n* = 38 for diameter of placentas, and *n* = 37 for weight of placentas for Saline, and *n* = 36 for length of fetuses, *n* = 36 for weight of fetuses, *n* = 37 for diameter of placentas, and *n* = 37 for weight of placentas for H1N1) and *Il1r2*−/− (*n* = 30 for length of fetuses, *n* = 30 for weight of fetuses, *n* = 30 for diameter of placentas, and *n* = 30 for weight of placentas for Saline, and *n* = 34 for length of fetuses, *n* = 34 for weight of fetuses, *n* = 34 for diameter of placentas, and *n* = 32 for weight of placentas for H1N1) mice (**g**). scale bar, 1 cm. **h**, **i** Representative images of H&E staining of vessels in decidua sections (**h**) and statistical analysis of vessel thickness (**i**) from pregnant *Il1r2*+/+ (*n* = 16 for Saline and *n* = 30 for H1N1) and *Il1r2*−/− (*n* = 20 for Saline and *n* = 6 for H1N1) mice. scale bar, 50 μm. **j** Representative images of immunohistochemistry staining of Cytokeratin in decidua sections from pregnant *Il1r2*+/+ and *Il1r2*−/− mice infected with either saline or H1N1. scale bar, 50 μm. The red line indicates the boundary between the maternal and fetal sides. The red star marks the site of trophoblast invasion into the decidua. M maternal, F fetal. Results are representative of two or three independent experiments. All bars in the graphs represent the mean ± s.e.m. Statistical comparisons were performed using one-way ANOVA with Tukey's multiple comparisons test (**c**, **e**, **g**, **i**). Source data are provided as a Source Data file.

matrix-remodeling factors, such as VEGF and MMP9, through the NF-κB and MAPK pathways[61,62]. Accordingly, elevated IL1R2 may inhibit these downstream pathways, reducing VEGF and MMP9 levels and impairing uterine remodeling—ultimately leading to fetal growth restriction. Targeting IL1R2, therefore, presents a promising strategy to mitigate the adverse reproductive consequences of respiratory viral infections.

Our findings, demonstrating that severe maternal pulmonary inflammation—rather than direct viral burden—drives adverse pregnancy outcomes in early gestation, may initially appear to contradict a recent study, which identified uncontrolled lung viral replication as the primary cause of fetal growth restriction in late-gestation maternal SARS-CoV-2-infected mice[8]. The differences in this mechanism observed at different stages of pregnancy might be due to the distinct roles of macrophages during the early and late stages of pregnancy. During early pregnancy, decidual macrophages are primarily tasked with immune tolerance and vascular remodeling (M2-like phenotype), essential for supporting placentation. In late pregnancy, the dominant physiological role of these macrophages shifts toward orchestrating the pro-inflammatory cascade required for parturition[63,64]. A systemic high-viral-load infection delivers a powerful inflammatory signal that prematurely activates or exacerbates this intrinsic parturition program, resulting in placental dysfunction and adverse outcomes.

Despite the compelling evidence presented, this study has several limitations. First, in this study, we employed a syngeneic mating model to clearly delineate the core pathway through which respiratory viral infection leads to fetal growth restriction, specifically the lung inflammation-systemic immune dysregulation-placental dysfunction axis. We fully acknowledge, as demonstrated by Engels et al., that allogeneic pregnancy models, which incorporate immune tolerance mechanisms toward the semi-allogeneic fetus and better recapitulate the immunological landscape of human pregnancy and hold significant translational value[65]. Consequently, we will employ allogeneic models in future studies to further validate the protective effects of DPP4 inhibition. Second, while the efficacy of the inhibitor has been demonstrated using the mouse-adapted influenza A/PR/8/34 (H1N1) strain, future studies should evaluate its therapeutic effect against more clinically relevant seasonal strains. Third, it is also important to note that while the natural transmission routes for MHV include fecal-oral and blood-borne pathways and not limited to the respiratory tract, differing from the primary respiratory transmission of SARS-CoV-2, intranasal inoculation of MHV establishes a robust infection in the lungs and elicits strong pulmonary immunopathology[66,67]. Therefore, the animal models used may not fully recapitulate the complex immunological and physiological interplay of human pregnancy and respiratory infection, underscoring the need for validation in clinical cohorts.

## Methods

### Mice

Seven-week-old female and eight-week-old male C57BL/6 mice were purchased from Shanghai SLAC Laboratory Animal Company (stock number: scb0105). *Il1r2*+/− mice were obtained from GemPharmatech and bred to generate *Il1r2*+/+ and *Il1r2*−/− genotypes (stock number: T005564). Genotyping was performed via standard PCR using primers listed in Supplementary Table 1. All mice were maintained on a C57BL/6 background. Littermate and cage-mate controls were used throughout the study. Mice were housed under specific pathogen-free (SPF) conditions, with a 12-h light/dark cycle, ambient temperature of 20–26 °C, and relative humidity of 50–70%. All animals including experimental and control mice were co-house throughout the study. All animals were fed an irradiation-sterilized mouse diet (WQ JXBIO-TECHNOLOGY). The established humane endpoints included: (1) weight loss exceeding 25% of initial body weight; (2) severe lethargy, hunched posture, or difficulty moving; (3) inability to access food or water independently; and (4) signs of significant distress such as dyspnea. Upon observation of any of these criteria, mice were immediately euthanized by $CO_2$ inhalation using a gradual fill method, followed by cervical dislocation for confirmation.

### Virus infections

The mouse-adapted influenza A/PR/8/34 (H1N1) strain was propagated in chicken embryonated eggs[68]. Murine hepatitis virus (MHV) was generously provided by X. Zhou (Wuhan Institute of Virology, University of Chinese Academy of Sciences)[69]. Female and male mice were paired in the late afternoon, and vaginal plug detection was performed the following morning; plug-positive mice were designated as embryonic day 0.5 (E0.5). At E6.5, pregnant mice were anesthetized and intranasally inoculated with a sublethal dose of 16 HA of PR8 in 50 μL sterile saline or MHV in 50 μL DMEM. Mice were monitored and weighed daily and euthanized at E12.5. At E12.5 after placentation, pregnant mice were anesthetized and intranasally inoculated with a sublethal dose of PR8 in 50 μL sterile saline. Mice were monitored and weighed daily and euthanized at E18.5. Lungs and uterine tissues were harvested, photographed, weighed, and either snap-frozen or fixed for further analyses. To assess the long-term impact of prenatal infection, pregnant mice were inoculated intranasally with PR8 at E6.5 and then permitted to deliver naturally. Maternal and offspring growth was tracked for 3 weeks postpartum, including weekly body weight assessments. All infection experiments were conducted under biosafety level 2 (BSL-2) conditions at the University of Science and Technology of China.

### qPCR

Total RNA was extracted from tissues or cells using TRIzol reagent (Invitrogen, 15596018). Reverse transcription was performed using a commercial kit (Accurate Biology, AG11705), and the resulting cDNA was

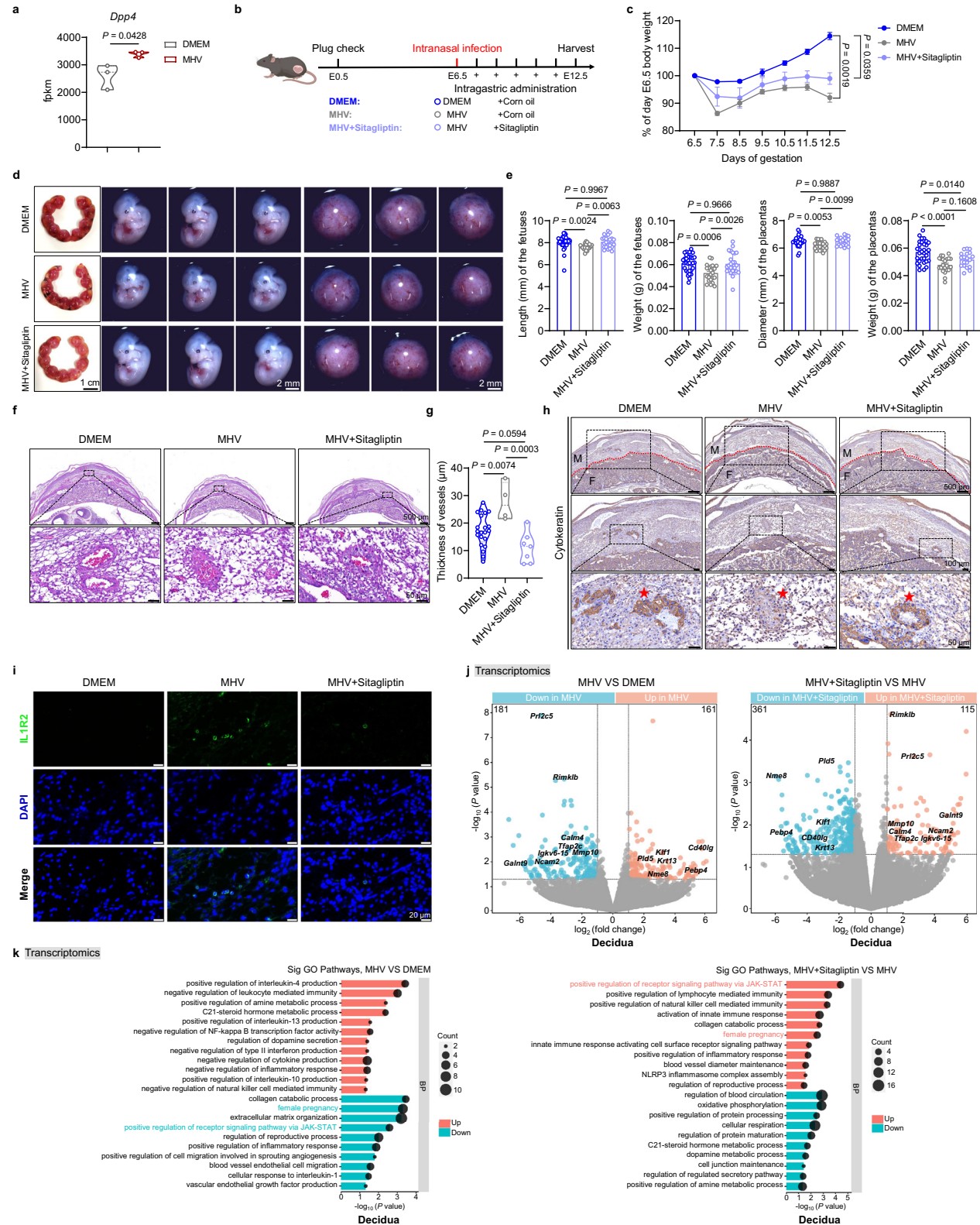

subjected to quantitative PCR (qPCR) using SYBR Green (Accurate Biology, AG11702). Primer sequences are listed in Supplementary Table 2. All primers were designed using Primer-BLAST and synthesized by Tsingke. Gene expression levels were quantified using the ΔCT method.

## Plaque assay

Viral titers in lung and decidual tissues harvested at E12.5 were determined by plaque assay on MDCK cells. Tissues were homogenized in ice-cold serum-free DMEM, and clarified supernatants were collected after centrifugation. MDCK monolayers in 24-well plates were inoculated with serial 10-fold dilutions of tissue supernatant for 2 h at 37 °C with intermittent agitation. Following adsorption, cells were overlaid with a semi-solid medium consisting of a 1:1 mixture of 2× DMEM (Procell system, PM150221) and 2% methylcellulose, supplemented with TPCK-trypsin (MedChemExpress, HY-129047C). Plates were then incubated for 2–4 days until plaque formation. Plaques were visualized

**Fig. 7 | Inhibition of DPP4 alleviates intrauterine growth restriction caused by pulmonary coronavirus infection modeled by intranasal MHV inoculation.** **a** Fpkm values of *Dpp4* expression in lungs infected with either DMEM ($n = 3$) or MHV ($n = 3$). **b** Schematic diagram illustrating the timeline of MHV infection and sitagliptin treatment of pregnant mice (Created in BioRender. Ding, X. (2026) https://BioRender.com/lfokc6w). **c** Changes in body weight of pregnant mice infected with either DMEM ($n = 4$) or MHV alone ($n = 3$), or treated with sitagliptin ($n = 3$). **d**–**k** Maternal deciduas were collected at embryonic day 12.5 (E12.5) for analysis. **d**, **e** Representative images of uterus, fetuses and placentas (**d**) and statistical analysis of the length and weight of fetuses and the diameter and weight of placentas (**e**) infected with either DMEM ($n = 41$ for length of fetuses, $n = 33$ for weight of fetuses, $n = 32$ for diameter of placentas, and $n = 32$ for weight of placentas) or MHV alone ($n = 24$ for length of fetuses, $n = 23$ for weight of fetuses, $n = 24$ for diameter of placentas, and $n = 22$ for weight of placentas), or treated with sitagliptin ($n = 24$ for length of fetuses, $n = 25$ for weight of fetuses, $n = 20$ for diameter of placentas, and $n = 19$ for weight of placentas). scale bar, 2 mm. **f**, **g** Representative images of H&E staining of vessels in decidua sections (**f**) and statistical analysis of the thickness of vessels (**g**) infected with either DMEM ($n = 21$) or MHV alone ($n = 4$), or treated with sitagliptin ($n = 7$). scale bar, 50 μm. **h** Representative images of immunohistochemistry staining of Cytokeratin in decidua sections infected with either DMEM or MHV

alone, or treated with sitagliptin. scale bar, 50 μm. The red line indicates the boundary between the maternal and fetal sides. The red star marks the site of trophoblast invasion into the decidua. M maternal, F fetal. **i** Representative images of immunofluorescence staining of IL1R2 in decidua sections infected with either DMEM or MHV alone, or treated with sitagliptin. scale bar, 20 μm. **j** Transcriptomic analysis of differentially expressed genes in deciduas infected with either DMEM ($n = 3$) or MHV alone ($n = 3$), or treated with sitagliptin ($n = 3$). **k** Gene Ontology (GO) of Biological Process (BP) analysis of differentially expressed genes in deciduas infected with either DMEM ($n = 3$) or MHV alone ($n = 3$), or treated with sitagliptin ($n = 3$). Results are representative of two or three independent experiments. All bars in the graphs represent the mean ± s.e.m. Statistical comparisons were performed using a two-tailed unpaired Student's $t$ test (**a**), one-way analysis of variance (ANOVA) with Dunnett's multiple comparisons test (**c**) and one-way ANOVA with Tukey's multiple comparisons test (**e**, **g**). Statistical significance of differential gene expression (**j**) was determined by the Wald test (two-sided) implemented in DESeq2, with $P$ values adjusted for multiple comparisons using the false discovery rate (FDR) method. Statistical significance of pathway enrichment (**k**) was determined by the hypergeometric test (one-sided), with $P$ values adjusted for multiple comparisons using the false discovery rate (FDR) method. Source data are provided as a Source Data file.

by formalin fixation and crystal violet staining. Viral titers were calculated from plaques counted in appropriate dilutions and expressed as PFU per gram of tissue.

## Leukocyte isolation

For lung leukocytes, lung tissue was minced and digested with 1 mg/mL collagenase type I (Sigma-Aldrich, C0130) in RPMI 1640 medium (Viva-Cell, Shanghai, C3018) at 37 °C for 1 h. Digests were filtered through nylon mesh, followed by density gradient centrifugation in 40% Percoll at $750 \times g$ for 20 min. Red blood cells were lysed, and leukocytes were collected. Mouse peripheral blood mononuclear cells were isolated after red blood cell lysis. For decidual leukocytes, tissues were harvested at E12.5 after embryos were carefully removed. The decidua was minced and digested in 2 mg/mL collagenase type IV (Sigma-Aldrich, C5138) in RPMI 1640 at 37 °C for 45 min. The resulting suspensions were filtered, and leukocytes were isolated using the same Percoll-based protocol.

## Flow cytometry analysis

Isolated cells were suspended in phosphate-buffered saline (PBS) containing 1% fetal bovine serum, blocked with Fc receptor antibodies, and stained with fluorophore-conjugated antibodies for 30 min at 4 °C in the dark. Cells were washed twice with PBS and analyzed using a BD Celesta flow cytometer. Data were processed with FlowJo V10. Gating strategies for lymphocyte and myeloid populations are shown in Supplementary Fig. 5. Antibodies used are listed in Supplementary Table 3.

## Histological and immunostaining analyses

Mouse lungs and decidual tissues were fixed overnight in 10% neutral-buffered formalin and embedded in paraffin following standard protocols. Sections were dewaxed with xylene, rehydrated through graded ethanol, and stained with hematoxylin and eosin (H&E). Masson's trichrome staining was performed by XL Pathology Tissue Services (Anhui, China). For immunohistochemistry (IHC) and immunofluorescence (IF), antigen retrieval was conducted by boiling sections for 15 min. Sections were incubated with primary antibodies and visualized using either a Biotin-Streptavidin Complex System (Zhongshan Golden Bridge, PV-9001 and PV-9002) or an Alexa Fluor Tyramide SuperBoost Kit (Thermo Fisher Scientific, B40922 and B40926), following manufacturers' protocols. Images were acquired using 3D HISTECH digital slide scanners and analyzed with Case Viewer software. Quantitative analysis of immunohistochemistry images was conducted using ImageJ software. Lung injury scoring was performed by examining ten randomly selected fields per sample. The scoring criteria were as follows: 0, normal lung with no inflammatory cells; 1,

mild inflammation with occasional inflammatory cells surrounding perivascular and peribronchial areas; 2, moderate inflammation with increased numbers of inflammatory cells in perivascular and peribronchial regions, along with some infiltration into the lung parenchyma; 3, severe inflammation with extensive inflammatory cell accumulation in perivascular and peribronchial areas and the formation of large inflammatory foci within the lung parenchyma. Antibodies used for IHC and IF are listed in Supplementary Table 4.

## In vivo drug administration

The DPP4 inhibitor sitagliptin (Selleck, S5079) was dissolved in DMSO and diluted in corn oil to a final concentration of 5.0 mg/mL, corresponding to a dose of 35–50 mg/kg/day. Pregnant mice received daily oral gavage of sitagliptin (200 μL) or vehicle control on E7.5, E8.5, E9.5, E10.5, and E11.5 when infected with virus at E6.5. And pregnant mice received daily oral gavage of sitagliptin (200 μL) or vehicle control on E13.5, E14.5, E15.5, E16.5, and E17.5 when infected with virus at E12.5.

## Cytokine antibody array

Lung and decidual tissues were collected from pregnant mice at E12.5 after intranasal administration of saline or H1N1 at E6.5. Cytokine profiles were assessed using the Mouse Cytokine Antibody Array (RayBiotech, QAM-TH17-1) according to the manufacturer's instructions. In brief: (i) 100 μL of sample diluent was added to each well to block the array surface for 30 min at room temperature; (ii) 100 μL of lysate (500 μg/mL) was added to each well and incubated overnight; (iii) biotinylated antibody cocktail was applied; (iv) Cy3-equivalent dye-conjugated streptavidin was used for detection. Fluorescence signals were scanned and analyzed using GenePix software.

## ELISA

Cytokine concentrations in tissue supernatants or serum were measured using enzyme-linked immunosorbent assay (ELISA) kits, following manufacturers' protocols. Kits used included IL1R2 (RayBiotech, ELM-IL1R2-1), VEGF (Multisciences, EK283), and DPP4 (Elabscience, E-EL-M3081).

## RNA-Seq and data analysis

Total RNA was extracted from mouse lung and decidual tissues using TRIzol reagent (Thermo Fisher Scientific). RNA quantity and integrity were evaluated with the RNA Nano 6000 Assay Kit on a Bioanalyzer 5400 system (Agilent Technologies, CA). Library preparation and sequencing were performed on the NovaSeq 6000 platform (Novogene, Beijing, China). After quality control, libraries were pooled based

on effective concentrations and expected data output and subjected to paired-end sequencing. Raw image files were processed with CASAVA software to generate sequence reads. Gene-level read counts were obtained using FeatureCounts (v2.0.6). Fragments per kilobase of exon per million mapped reads (FPKM) were calculated for each gene, normalized by transcript length and total mapped reads. Differential gene expression analysis among three experimental groups (each with three biological replicates) was conducted using DESeq2 (v1.42.0). Functional enrichment of differentially expressed genes (DEGs) was assessed using clusterProfiler (v4.8.1) for Gene Ontology (GO) terms. Enrichment analysis was also performed using the Reactome pathway database, with pathways considered significantly enriched at adjusted $p$ values < 0.05.

## Proteomic analysis
At embryonic day 12.5 (E12.5), pregnant mice were euthanized for the collection of serum and decidual tissues. Protein extraction, enzymatic digestion, LC–MS/MS analysis, and data processing were carried out by iProteome Biotechnology (Shanghai, China). Briefly, tissue and serum samples were homogenized in 100 μL of 50 mM ammonium bicarbonate (ABC) buffer, and proteins were denatured at 95 °C for 5 min. After cooling, proteins were digested overnight at 37 °C with trypsin at a 1:25 enzyme-to-protein ratio. Samples were analyzed using an EASY-nLC 1200 system coupled to a Q Exactive HF-X Hybrid Quadrupole-Orbitrap mass spectrometer (Thermo Fisher Scientific) via a nano-electrospray ion source. Mass spectrometry was performed in data-independent acquisition (DIA) mode. An initial full MS1 scan (m/z 300–1400) was acquired at a resolution of 30,000, with an AGC target of 3E + 06 and maximum injection time of 20 ms. This was followed by 30 DIA segments at a resolution of 15,000, with AGC target set to 1E + 06 and maximum injection time of 22 ms. Peptide identification and protein quantification were performed using the Firmiana platform (https://phenomics.fudan.edu.cn/firmiana/). Raw files were searched against the UniProt mouse proteome using FragPipe (v12.1) integrated with MSFragger (v2.2). DIA data were further analyzed using DIA-NN (v1.7.0), with peptide quantification based on the mean chromatographic fragment ion peak areas across all reference spectral libraries. Label-free quantification was applied to determine protein abundance.

## Cell sorting and single-cell RNA sequencing
Mononuclear cells were isolated from lung and decidual tissues at E12.5 and stained with an anti-CD45 antibody. CD45-positive cells were sorted using a Thermo Fisher Scientific Bigfoot Full-Spectrum Flow Cytometer and processed for single-cell RNA sequencing. Cell viability was assessed microscopically using Trypan Blue staining. Single-cell suspensions were prepared at a concentration of $2 \times 10^5$ cells/mL in PBS (HyClone) and loaded onto a microwell chip using the Singleron Matrix NEO™ Single Cell Processing System. Barcoding Beads were collected from the chip, followed by reverse transcription of captured mRNA and cDNA synthesis via PCR amplification. The amplified cDNA was fragmented and ligated with sequencing adapters. scRNA-seq libraries were constructed using the GEXSCOPE® Single Cell RNA Library Kits (Singleron). Individual libraries were diluted to 4 nM, pooled, and sequenced on an Illumina NovaSeq 6000 system with 150 bp paired-end reads.

Raw scRNA-seq reads were processed into gene expression matrices using the CeleScope pipeline (v1.9.0; https://github.com/singleron-RD/CeleScope). Briefly, low-quality reads were removed, and poly-A tails as well as adapter sequences were trimmed using Cutadapt (v1.17). Cell barcodes and UMIs were extracted, and reads were aligned to the GRCm38 reference genome (Ensembl release 92) using STAR (v2.6.1a). UMI and gene counts per cell were quantified with feature-Counts (v2.0.1), and expression matrices were generated for downstream analysis.

## scRNA-seq data analysis
Count matrices were imported and analyzed in Python (v3.9) using the Scanpy package (v1.10.0). Cells with fewer than 200 detected genes or genes expressed in fewer than three cells were excluded. Cells exhibiting high mitochondrial gene content (>15%) were also removed. Potential doublets were identified and filtered out using Scrublet, applied per sample. The filtered data were normalized to 10,000 counts per cell and log1p-transformed. Highly variable genes were selected with batch key specification, and batch effects were corrected using Harmony integration within the principal component analysis (PCA) space. Corrected embeddings were used to construct a neighborhood graph, followed by UMAP visualization and clustering via the Leiden algorithm at an appropriate resolution.

For subpopulation analysis, cells previously annotated as macrophages were subsetted from the integrated dataset. These cells were then re-embedded, re-clustered, and re-analyzed independently to characterize distinct subtypes and functional states.

## Statistical analysis
Statistical analysis was performed using GraphPad Prism 8.0 (GraphPad Software, San Diego, CA, USA). No statistical method was used to predetermine sample size. In vivo experiments were randomized across treatment groups, and no data were excluded from analysis. Comparisons between two groups were evaluated using unpaired two-tailed Student's $t$-tests, while comparisons among multiple groups were analyzed by one-way ANOVA. A $P$ value < 0.05 was considered statistically significant. All data are expressed as mean ± standard error of the mean (s.e.m). Details of statistical methods are provided in the respective figure legends.

## Ethics statement
All research complied with relevant ethical regulations. The study protocol was approved by the guidelines of Animal Protection Law and the approved protocols by the Ethics Committee of the University of Science and Technology of China (approval number: USTCA-CUC27120122083) and adhered to the National Animal Research Guidelines (China).

## Reporting summary
Further information on research design is available in the Nature Portfolio Reporting Summary linked to this article.

# Data availability
The raw sequencing data generated in this study have been deposited in the Genome Sequence Archive (GSA) at the National Genomics Data Center under accession number CRA023974 and are publicly accessible at https://ngdc.cncb.ac.cn/gsa/browse/CRA023974. The proteomics data generated in this study have been deposited in the OMIX repository at the National Genomics Data Center under accession number OMIX013325. Please note that registration is required via the NGDC platform to access the raw data files. All data are included in the Supplementary Information or available from the authors, as are unique reagents used in this Article. The raw numbers for charts and graphs are available in the Source Data file whenever possible. Source data are provided with this paper.

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

## Acknowledgements

This work was supported by the National Key Research & Developmental Program of China (2022YFC2702400, B.F.; 2025YFC2708002, B.F.; 2022YFA1103603, H.W.); the National Natural Science Foundation of China (32270979, B.F.; 32170932, Y.Z.); the Strategic Priority Research Program of the Chinese Academy of Sciences (XDB0940201, B.F.); the Global Select Project of the Institute of Health and Medicine, Hefei Comprehensive National Science Center (DJK-Lx-2022004, B.F.); USTC Research Funds of the Double First-Class Initiative (YD9100002030, B.F.); Youth Innovation Promotion Association of Chinese Academy of Sciences (Y2023127, B.F.).

## Author contributions

G.S. designed and performed the experiments, and analyzed and interpreted the data. S.X. and M.L. contributed to the development of mouse models. Y.C. contributed to the analysis of RNA-seq data. Y.Z. assisted with background research. H.W. provided strategic planning. B.F. supervised the project, provided crucial ideas, and assisted with data interpretation. B.F. wrote the manuscript with G.S.

## Competing interests

The authors declare no competing interests.
