## [Transparent Peer Review File · Nature Communications]

Dipeptidylpeptidase 4 inhibition attenuates gestational pathologies via immune homeostasis restoration in the pulmonary-uterine axis

Corresponding Author: Professor Binqing Fu

Version 0:

Reviewer comments:

Reviewer #1

(Remarks to the Author)

This is an exciting series of studies highlighting a mechanism by which respiratory viruses can impact placenta vasculature and intrauterine growth restriction of fetuses. Identification of DPP4 and IL-1 family genes as mediators is novel and exciting because it offers therapeutic potential. My specific comments are below:

1. Virus titers: the authors must measure virus titers in respiratory tissues. While there are ample studies illustrating that neither influenza viruses nor coronaviruses replicate in reproductive tissues in mice, whether viral loads are reduced with the DPP4 or IL1R2 manipulations must be shown and cannot just be inferred from transcriptional profiling showing changes in type I IFN signaling.
2. Rational for infecting prior to placentation: the authors should provide a rationale for only testing infection during a single first trimester human equivalent time. Would these treatments work equally well if infections occurred after embryonic day 10.5, after the placenta is formed?
3. Immunophenotype: with a distinct focus on mucosal immunity (e.g., that is the focus of the Discussion), it was surprising that immune cell frequencies were only measured in peripheral blood and not in the lungs and placenta. This seems to be a missing link in showing that the mucosal immune profile is altered.
4. Coronavirus data are confusing: because virus titers are not measured, it is difficult to interpret data when previous work with SARS-Cov-2 infection in dams, cited by the authors, showed that reduced control of virus replication and dampened type I IFN responses in the lungs, more than placental inflammation, caused intrauterine growth restriction, which was reversed by antiviral treatment. With MHV, the authors claim that the opposite is true but show no virus replication data. It also is concerning that MERS uses DPP4 as a virus entry receptor and it is not clear if MHV does as well. Thus, DPP4 could be acting differently with MHV than influenza infection. This is never evaluated or discussed. The authors assume that DPP4 effects are identical for both viruses, which is confusing when in Fig 7e it is apparent that sitagliptin has no effect on MHV induced intrauterine growth restriction, yet the authors conclude that sitagliptin reverses the effects of MHV on offspring. Thus, this suggests that DPP4 is not the mechanism of action for MHV.
5. Minor: Fig 3f is not mentioned in text
6. Minor: there is no discussion of the limitations of this study.

Reviewer #2

(Remarks to the Author)

The manuscript by Shi et al. attempts to unravel how lung infections might disrupt immune responses necessary for vascular remodeling and trophoblast invasion and ultimately the health of the embryos. They nicely show that respiratory infection with flu or MHV-1 enhances Cd11b+ IL1R2 expressing cells that accumulate in the uterus and the embryos show evidence of growth restriction. Importantly, a DPP4 inhibitor could reverse the detrimental effects of pulmonary infection on embryo health. The paper is very interesting and well done, but there is still confusion over how the IL1R2 is accumulating in the uterus as noted below.

Major

- 1) Line 76 and again line 211 asserts that the IL1R2 accumulation is via soluble protein (but this is not proven in this

manuscript). In reality, is the accumulation of IL1R2 in the uterus due to soluble protein spilling from the lung or Cd11b+ cell recruitment to the uterus which express IL1R2? Figure 3j seems to show all of the IL-1R2 as cell-associated. Fig 4f and g show CD11b+ cells that are IL1R2+ in the uterus, so it appears that the increase is associated with cellular influx in the uterus and this is how the discussion is framed as well. However, if this is true, it is hard to explain why the mRNA expression of IL1R2 in the decidua is not changed (Fig 3f), but the protein is higher (Fig 3g). If the increased myeloid cells bring the IL-1R2 to the uterus, shouldn't you see evidence of increased mRNA by virtue of increased cell number? This is even more confusing when looking at the MHV data in Fig 6 m where MHV upregulates IL1R2 mRNA in the lung but not protein (so there is likely no increased IL1R2 protein to leak from lung in this case), and in the decidua, MHV elevates IL1R2 protein again without showing evidence of increased mRNA. The authors need to clarify how the elevated IL1R2 protein levels are happening. Perhaps the post-transcriptional regulation of this protein is important.

2) The authors seem to imply that the increased percentages of myeloid cells may be responsible for the IL1R2 levels in the uterus. The data are presented only as percentages which implies that if the CD11b+ percentage is up, something should be lower, but that is not shown. Also, how do the actual numbers of cells change over time post-infection? Knowing this may help to clarify the source of the IL1R2 that accumulates in the uterus.

3) Do you see the same impacts on the uterus if myeloid cell migration is blocked post lung infection (e.g. use of CCR2 KO mice, i.v. clodronate depletion of monocytes)? This could really help clarify whether it is cellular trafficking that is essential.

4) Does the growth restriction of the fetus carry forward to smaller pups when born? If you let the infected mice recover and give birth, would the neonates show defects in growth or health?

Minor:

1) What is the cellular source of the DPP4 production?

2) Supplemental Fig 3g purports to measure viral H1N1 protein, but in reality this is mRNA for a viral protein measured by RT-PCR it seems. The methods and text do not state what protein is measured, and for influenza, viral RNA is a poor indicator of infectious virus. The authors should determine infectious titer of flu in lung with and without the DPP4 inhibitor.

3) What does the red font indicate in Figures 1c and d?

4) What was the sublethal dose of PR8 used?

5) The mechanistic link between how DPP4 inhibition limits IL1R2 accumulation in the uterus or even the lung is not defined, or speculated upon. The authors admit this in the discussion, but a hypothesis would be interesting to speculate upon.

Reviewer #3

(Remarks to the Author)

The authors show that gestational influenza leads to immune activation in the uterus defined by activation of inflammatory cytokines leading to intrauterine growth restriction in mice. Using transcriptomic analysis the authors show that systemic DPP4 upregulation (lung, decidua and blood) is associated with adverse outcome. Inhibition of DPP4 with Sitagliptin improves adverse outcome as defined by reduced inflammation and reduced growth restriction. This is an important study elucidating key mechanisms involved in lung infection mediated complications during pregnancy. The methodology used is sound and the conclusions drawn mostly supported by the data. However, there are major concerns that dampen my enthusiasm.

Major comments

#1: DPP4 is a known protein associated with signal transduction, immune regulation and glucose metabolism (including gestational diabetes). In all the animal experiment that were involving Sitagliptin treatments, the authors did not include the control with the treatment alone. Since Sitagliptin has per se an effect on the immune function would have been important to compare the H1N1+ Sitagliptin group vs Sitagliptin, instead of only not infected. This is a major drawback of the study. These experiments are essential to support the major conclusion of the manuscript.

#2: The authors state (lines 240-243): "Il1r2 mRNA was strongly induced in the lung but not in the uterus (Fig. 3c, f), these results suggest that respiratory virus infection induces IL1R2 expression in lung-resident myeloid cells, which then disseminate through the bloodstream and accumulate systemically, including at the maternal-fetal interface."

In order to evaluate and distinguish the different subsets of myeloid cells, it would be important to include some key markers that better characterize resident and infiltrating cells. The macrophages subsets in the decidua are characterized not only by the expression of CD11b and F4/80 (classical macrophages markers), but also by CD11c, CX3CR1, TREM2 and a low expression of Ly6C, while infiltrating macrophages travelling via the bloodstream express CCR2.

#3: Line 296, the authors state: "These findings parallel the immunological alterations observed in influenza infection and support the notion that respiratory viral infections impair placental function and fetal development via indirect, non-viral mechanisms."

Although MHV can be used as surrogate model for SARS-CoV-2 pathogenesis, it is not a respiratory borne virus, spreading mainly via fecal-oral way or blood transmission. I would suggest to completely rephrase the section on MHV, including the title, since MHV can not be compared to all respiratory viral infection, although most clinical symptoms are similar. The same counts for the abstract. Most respiratory viruses, including SARS-CoV, have a marked tropism for ciliated cells and they are mainly localized in the respiratory tract, while the infection with MHV is multi-organ. Thus, the statements made on respiratory viruses by including MHV data need to be revised throughout the manuscript accordingly.

Minor comments:

#1: In Fig. 2d the significance regarding the H1N1 vs H1N1-Sigtaglipin group is missing (as mentioned in the text line 134-135). Please amend.

#2: A minor comment regarding the gating strategy in Suppl. Fig. 4 A: the gate on CD45+ cells is set with living cells. There was a pre-gating strategy not shown with a marker for living cells (e.g. live/dead) otherwise looks like a gate towards a physical parameter. Please clarify.

#3: The authors use the mouse-adapted influenza A/PR/8/34 (H1N1). Would have been interesting to see the effect of the treatment on a more clinically relevant seasonal strain. Please discuss.

#4: In order to compare the percentages during steady state control and infection the absolute number of cells should have been recorded. During infection, normally, a high amount of cells are recruited and activated and there is an increase in the total amount of CD45+ cells that influence the relative percentage number. Both representative gating strategy for saline and H1N1 condition should be shown.

#5: In il1r2-KO model would have been interesting to see how and if the immune cell composition change in relation to the migration from the lung to the placenta. Please discuss.

#6: the authors should address why they have not used the allogenic mouse model that was shown by various groups to mimic best infection-immunity aspects during human pregnancy (Engels et al., Cell Host and Microbe 2017). Please discuss and cite accordingly.

#7: in the methods section, the humane endpoint used is not mentioned. Please amend.

Reviewer #4

(Remarks to the Author)

Version 1:

Reviewer comments:

Reviewer #1

(Remarks to the Author)

The authors did a very thorough job responding to our concerns and provide substantial evidence to support claims in the manuscript. We have no additional concerns or comments.

Reviewer #2

(Remarks to the Author)

The revisions have strengthened the manuscript and conclusions and addressed my concerns.

Reviewer #3 and Reviewer #4 (Remarks to the Author):

Inhibition of DPP4 alleviates intrauterine growth restriction caused by respiratory influenza virus infection The authors have adequately addressed all major and minor issues. The addition of requested internal controls as well as the analysis of the different cell subsets strengthened the conclusions drawn an the manuscript has significantly improved. However, one minor issue remains. The analysis of the cell composition in the lung after Sigagliptin administration should have included internal controls as well for better comparability and validation of the data. It would be great if the authors could include this analysis as well to further strengthen the effectivity of their drug candidate.

We sincerely thank the Reviewers and the Editor for their thoughtful and constructive comments. We have carefully revised the manuscript accordingly and performed additional experiments as requested. Below, we provide a point-by-point response to all the reviewers' comments. Reviewer comments are shown in Blue color, and our responses follow each comment.

REVIEWER COMMENTS

Reviewer #1 (Remarks to the Author):

This is an exciting series of studies highlighting a mechanism by which respiratory viruses can impact placenta vasculature and intrauterine growth restriction of fetuses. Identification of DPP4 and IL-1 family genes as mediators is novel and exciting because it offers therapeutic potential. My specific comments are below:

Reply: We really appreciate your comments.

1. Virus titers: the authors must measure virus titers in respiratory tissues. While are ample studies illustrating that neither influenza viruses nor coronaviruses replicate in reproductive tissues in mice, whether viral loads are reduced with the DPP4 or IL1R2 manipulations must be shown and cannot just be inferred from transcriptional profiling showing changes in type I IFN signaling.

Reply: Thank you for pointing out the virus titers shortage in the 1st edition of our manuscript. To address your concern, we have conducted additional experiments to assess virus titers in both the DPP4 or IL1R2 treatment.

In the influenza infection model, we systematically evaluated viral replication and distribution using multiple approaches: (1) Immunohistochemistry for influenza nucleoprotein revealed positive signals exclusively in infected lungs, with no detection in the decidua. (2) Plaque assay quantification confirmed that infectious virus was only detectable in lung homogenates, and DPP4 inhibitor treatment did not reduce pulmonary viral titers. (3) qPCR for viral gene replication yielded consistent results, showing viral replication confined to the lungs, unaffected by sitagliptin.

Revised Supplementary Figure 2. Influenza viral replication in the lung and decidua following sitagliptin treatment.

In the *Il1r2* knockout model, immunohistochemistry and qPCR similarly confirmed that viral infection was strictly restricted to the lungs, and *Il1r2* knockout did not alter viral replication or tissue distribution.

Revised Supplementary Figure 8. Influenza viral replication in the lung and decidua following knocking out *Il1r2*.

In the coronavirus (MHV) infection model, qPCR analysis demonstrated that viral replication was likewise confined to the lungs and was not reduced by DPP4 inhibitor treatment.

Revised Supplementary Figure 10i. Intranasal MHV inoculation results in the accumulation of IL1R2 and inhibition of DPP4 alleviates lung inflammation.

These comprehensive virological data have now been included as supplementary figures. They collectively provide strong evidence supporting our central conclusion that respiratory viral infections are strictly confined to the maternal lungs without dissemination to the maternal-fetal interface, and the protective effects of DPP4 inhibition or *Il1r2* deficiency are mediated by modulating the host's immunopathological response, not by direct antiviral activity. We thank the reviewer for helping us strengthen the evidence in our paper.

2. Rational for infecting prior to placentation: the authors should provide a rationale for only testing infection during a single first trimester human equivalent time. Would these treatments work equally well if infections occurred after embryonic day 10.5, after the placenta is formed?

Reply: Thank you for bring this important point out. According to your suggestions, we infected pregnant mice were intranasally infected with influenza virus after the establishment of the choriovitelline placenta (at gestational day 12.5, E12.5)¹. This was followed by a 5-day treatment regimen with sitagliptin administered daily by oral gavage from E13.5 to E17.5, while control groups received the vehicle only. Our data showed that sitagliptin treatment still significantly mitigated virus-induced maternal weight loss and improved pregnancy outcomes. Treated embryos displayed increased weight and body length, compared to untreated infected controls (**revised 2nd edition Supplementary Figure 4**). These data suggest that DPP4 inhibitor sitagliptin treatments work equally well even infections occurred after embryonic day 12.5, after the placenta is formed.

Revised Supplementary Figure 4. Inhibition of DPP4 also alleviates intrauterine growth restriction caused by respiratory influenza virus infection after placentation.

3. Immunophenotype: with a distinct focus on mucosal immunity (e.g., that is the focus of the Discussion), it was surprising that immune cell frequencies were only measured in peripheral blood and not in the lungs and placenta. This seems to be a missing link in showing that the mucosal immune profile is altered.

Reply: Thank you for this suggestion. In the revised 2nd edition, we repeated the H1N1 infection model and analyzed immune cell frequencies in the lungs, peripheral blood, and

decidua. Our data showed that Infection induced a robust expansion of CD11b⁺ myeloid cells (macrophages, neutrophils and monocytes) in the lung, accompanied by a similar increase in circulating percentage of neutrophils and monocytes. By contrary, we did not observe expansion of CD11b⁺ myeloid cells in the decidua (**revised 2nd edition Supplementary Figure 6**), suggesting that IL1R2⁺ myeloid cells subset rather than the whole myeloid cells increased in decidua.

Revised Supplementary Figure 6. Differential alterations in myeloid cell populations in maternal lungs, peripheral blood and deciduas following influenza infection.

4. Coronavirus data are confusing: because virus titers are not measured, it is difficult to interpret data when previous work with SARS-Cov-2 infection in dams, cited by the authors, showed that reduced control of virus replication and dampened type I IFN responses in the

lungs, more than placental inflammation, caused intrauterine growth restriction, which was reversed by antiviral treatment. With MHV, the authors claim that the opposite is true but show no virus replication data. It also is concerning that MERS uses DPP4 as a virus entry receptor and it is not clear if MHV does as well. Thus, DPP4 could be acting differently with MHV than influenza infection. This is never evaluated or discussed. The authors assume that DPP4 effects are identical for both viruses, which is confusing when in Fig 7e it is apparent that sitagliptin has no effect on MHV induced intrauterine growth restriction, yet the authors conclude that sitagliptin reverses the effects of MHV on offspring. Thus, this suggests that DPP4 is not the mechanism of action for MHV.

Reply: We apologize for this confusion. We have substantially revised the text to address these concerns:

1). Viral Replication and Mechanism: We acknowledge that the absence of viral replication data in the initial submission was an oversight. In the revised version, we clearly state that MHV replication is confined to the lungs with no vertical transmission and have included the critical new experimental data showing that sitagliptin treatment does not affect *MHV* replication in the lungs. This supports our conclusion that the benefits of sitagliptin stem from its modulation of host immunopathology, not a direct antiviral effect.

Revised Supplementary Figure 10i. Intranasal MHV inoculation results in the accumulation of IL1R2 and inhibition of DPP4 alleviates lung inflammation.

2). Thank you for mention the previous work with SARS-Cov-2 infection in dams². They showed that reduced control of virus replication and dampened type I IFN responses in the lungs, more than placental inflammation, caused intrauterine growth restriction, which was reversed by antiviral treatment. Importantly, they indicated that infection at E16 (third trimester equivalent) rather than infection at either E6 (first trimester equivalent) or E10 (second trimester equivalent) has the above phenomena. Our study demonstrates that in an early-gestation model with a virus causing localized severe pneumonia, the magnitude of the maternal pulmonary inflammatory response is the primary driver. The differences in this mechanism observed at different stages of pregnancy might be due to the distinct roles of macrophages during the early and late stages of pregnancy.

During early pregnancy, decidual macrophages are primarily tasked with immune tolerance

and vascular remodeling (M2-like phenotype), essential for supporting placentation. A severe maternal inflammatory response can remotely disrupt this constructive program, leading to impaired placental development and fetal growth restriction. In late pregnancy, the dominant physiological role of these macrophages shifts toward orchestrating the pro-inflammatory cascade required for parturition. A systemic high-viral-load infection (e.g., the maSCV2 model in the JCI study) delivers a powerful inflammatory signal that prematurely activates or exacerbates this intrinsic parturition program, resulting in placental dysfunction and adverse outcomes.

We have added these important discussions in the revised manuscript (**Page 12-13 and Line 502-512**): “Our findings, demonstrating that severe maternal pulmonary inflammation—rather than direct viral burden—drives adverse pregnancy outcomes in early gestation, may initially appear to contradict a recent study, which identified uncontrolled lung viral replication as the primary cause of fetal growth restriction in late-gestation maSCV2-infected mice². The differences in this mechanism observed at different stages of pregnancy might be due to the distinct roles of macrophages during the early and late stages of pregnancy. During early pregnancy, decidual macrophages are primarily tasked with immune tolerance and vascular remodeling (M2-like phenotype), essential for supporting placentation. In late pregnancy, the dominant physiological role of these macrophages shifts toward orchestrating the pro-inflammatory cascade required for parturition^{3,4}. A systemic high-viral-load infection delivers a powerful inflammatory signal that prematurely activates or exacerbates this intrinsic parturition program, resulting in placental dysfunction and adverse outcomes.”

3). Mechanism of DPP4 Action: We appreciate the reviewer’s insightful question regarding viral entry receptors. It is well established that MERS-CoV uniquely utilizes DPP4 as its primary entry receptor, which is a key determinant of its distinct pathogenesis. In contrast, the MHV strain employed in our study utilizes CEACAM1^{5,6} as its canonical and necessary receptor for infection in mice, not DPP4. Therefore, our data indicate that in both influenza and MHV models, DPP4 inhibition alleviates fetal growth restriction by modulating the immune response, reducing pulmonary inflammation, and improving placental function, independently of suppressing viral replication. This suggests a key role for DPP4 in a common immunopathological pathway triggered by respiratory viral infections. We added these as follows in the Result (**Page 10 and Line 416-423**): “To further directly evaluate whether sitagliptin exerts its beneficial effects via antiviral activity in the MHV infection model, we measured viral gene replication in lung tissues after treatment. Critically, sitagliptin did not suppress *MHV* replication in the lungs and *MHV* remained restricted to the lung without spreading to the maternal-fetal interface (Supplementary Fig. 10i), suggesting that DPP4 may not be involved in MHV amplification and that the receptor for MHV remains CEACAM1, as previously reported⁶. These data clearly demonstrate that the protective effect of sitagliptin on pregnancy outcomes is not mediated by direct antiviral mechanisms, but rather through modulation of the host’s immunopathological response.”

4). Interpretation of Fig 7e Data: We have re-examined the data in Fig 7e that sitagliptin treatment significantly reversed MHV-induced fetal growth restriction (reflected in embryo weight, length, and placental size), with conclusions consistent with the data presented.

Revised Figure 7. Inhibition of DPP4 alleviates intrauterine growth restriction caused by respiratory coronavirus infection.

5. Minor: Fig 3f is not mentioned in text.

Reply: Sorry for this problem. It has been well taken in the revised 2nd edition of manuscript.

6. Minor: there is no discussion of the limitations of this study.

Reply: Thank you for this suggestion. We have added limitation discussion as follows (**Page 12-13 and Line 513-528**): “Despite the compelling evidence presented, this study has several limitations. First, in this study, we employed a syngeneic mating model to clearly delineate the core pathway through which respiratory viral infection leads to fetal growth restriction, specifically the lung inflammation-systemic immune dysregulation-placental dysfunction axis. We fully acknowledge, as demonstrated by Engels et al., that allogeneic pregnancy models, which incorporate immune tolerance mechanisms toward the semi-allogeneic fetus and better recapitulate the immunological landscape of human pregnancy and hold significant translational value⁷. Consequently, we will employ allogeneic models in future studies to further validate the protective effects of DPP4 inhibition. Second, while the efficacy of the inhibitor has been demonstrated using the mouse-adapted influenza A/PR/8/34 (H1N1) strain, future studies should evaluate its therapeutic effect against more clinically relevant seasonal strains. Third, it is also important to note that while the natural transmission routes for MHV include fecal-oral and blood-borne pathways and not limited to the respiratory tract, differing from the primary respiratory transmission of SARS-CoV-2, intranasal inoculation of MHV establishes a robust infection in the lungs and elicits strong pulmonary immunopathology^{8,9}. Therefore, the animal models used may not fully recapitulate the complex immunological and physiological interplay of human pregnancy and respiratory infection, underscoring the need for validation in clinical cohorts.”

Reviewer #2 (Remarks to the Author):

The manuscript by Shi et al. attempts to unravel how lung infections might disrupt immune responses necessary for vascular remodeling and trophoblast invasion and ultimately the health of the embryos. They nicely show that respiratory infection with flu or MHV-1 enhances Cd11b⁺ IL1R2 expressing cells that accumulate in the uterus and the embryos show evidence of growth restriction. Importantly, a DPP4 inhibitor could reverse the detrimental effects of pulmonary infection on embryo health. The paper is very interesting and well done, but there is still confusion over how the IL1R2 is accumulating in the uterus as noted below.

Reply: We really appreciate your comments.

Major:

1) Line 76 and again line 211 asserts that the IL1R2 accumulation is via soluble protein (but this is not proven in this manuscript). In reality, is the accumulation of IL1R2 in the uterus due to soluble protein spilling from the lung or Cd11b⁺ cell recruitment to the uterus which express IL1R2? Figure 3j seems to show all of the IL-1R2 as cell-associated. Fig 4f and g show CD11b⁺ cells that are IL1R2⁺ in the uterus, so it appears that the increase is associated with cellular influx in the uterus and this is how the discussion is framed as well. However, if this is true, it is hard to explain why the mRNA expression of IL1R2 in the decidua is not changed (Fig 3f), but the protein is higher (Fig 3g). If the increased myeloid cells bring the IL-1R2 to the uterus, shouldn't you see evidence of increased mRNA by virtue of increased cell number? This is even more confusing when looking at the MHV data in Fig 6 m where MHV upregulates IL1R2 mRNA in the lung but not protein (so there is likely no increased IL1R2 protein to leak from lung in this case), and in the decidua, MHV elevates IL1R2 protein again without showing evidence of increased mRNA. The authors need to clarify how the elevated IL1R2 protein levels are happening. Perhaps the post-transcriptional regulation of this protein is important.

Reply: Thank you very much for pointing out this important point. As you have mentioned, we observed evidence that the IL1R2 is cell-associated, especially myeloid cells. However, we previously did not find mRNA level of IL1R2 significantly increased in decidua. According to your suggestions, we have reflected on our experimental methods and speculate that it might be because the samples we used for the PCR tests were the whole decidua tissues, and the IL1R2⁺ immune cells accounted for a very limited small proportion in the entire decidua tissues. If we sort the CD45⁺ immune cells from the lung and decidua tissues for the experiment, it is possible to observe the transcriptional changes of IL1R2⁺ immune cells in the lungs and uterus (**revised 2nd edition Supplementary Figure 7**).

Therefore, in the revised manuscript, we analyzed single-cell RNA sequencing of CD45⁺ cells in the lung and uterus from dams at E12.5 following treatment with saline, H1N1 infection alone, or H1N1 infection plus sitagliptin. Our data showed that macrophages constituted the most abundant subset and were significantly expanded following H1N1 infection, an effect that was substantially reversed by sitagliptin treatment. Further analysis revealed a marked H1N1-induced upregulation of *Il1r2* in lung macrophages, which was also significantly suppressed by

sitagliptin. In the decidua, we identified ten immune cell clusters. Analysis of the major macrophage population showed a congruent pattern: H1N1 infection robustly induced *Il1r2* expression, and this induction was attenuated by sitagliptin. These findings underscore macrophages as a key *Il1r2*-expressing population during influenza infection in pregnancy.

Revised Supplementary Figure 7. A single-cell atlas of the maternal immune response to influenza infection in the lung and decidua.

We next re-clustered macrophages and searched for the most relevant macrophage subpopulations. A subpopulation of immunoregulatory macrophages highly expressed *Il1r2* was rare under homeostatic conditions but expanded significantly after H1N1 infection, with its proportion subsequently normalized by sitagliptin. Visualization of *Il1r2* expression confirmed its specific enrichment in this immunoregulatory subset after infection, which was

abrogated by treatment, suggesting these cells may serve as a key target of sitagliptin and represent a primary source of IL1R2. An analogous immunoregulatory macrophage subset, also exhibiting high *Il1r2* expression, was identified in the decidua. Its abundance increased after infection and decreased with sitagliptin treatment, and its *Il1r2* expression mirrored this trend, closely recapitulating the pulmonary response. This suggests that these cells may serve as a critical link between the lung and maternal-fetal interface, and constitute a local source of IL1R2 (revised 2nd edition Figure 5).

Revised Figure 5. Inhibition of DPP4 reduces migratory and *Il1r2*-high macrophages following influenza virus infection.

We also discuss this important point in the discussion part as follows (**Page 12 and Line 481-492**): “Notably, in both influenza and coronavirus infection models, uterine IL1R2 protein levels increased without corresponding detection in virus titers or IFN β connected genes expression. Conversely, in lung tissue, viral infection led to elevated *Il1r2* mRNA expression and significantly increased of IL1R2⁺ myeloid cells, implying that IL1R2 accumulates in the uterus via systemic circulation rather than *in situ* production. Flow cytometric analyses further confirmed increased IL1R2 expression on myeloid cells within the lung, peripheral blood, and decidua following infection. The gradient—highest in the lung, intermediate in blood, and lowest in the decidua—suggests a migration trajectory of IL1R2-high cells from the lung to the uterus. Our scRNA data further identifies tissue-homing IL1R2⁺ macrophages that link pulmonary infection to maternal-fetal interface. Together, these findings support a model in which IL1R2 functions as a molecular conduit linking the immune landscapes of the respiratory and reproductive tracts, reinforcing and refining the concept of the common mucosal immune system.”

2) The authors seem to imply that the increased percentages of myeloid cells may be responsible for the IL1R2 levels in the uterus. The data are presented only as percentages which implies that if the CD11b⁺ percentage is up, something should be lower, but that is not shown. Also, how do the actual numbers of cells change over time post-infection? Knowing this may help to clarify the source of the IL1R2 that accumulates in the uterus.

Reply: Thank you for bring this important point up. First, our single-cell data show that pulmonary macrophages are the cell type with the greatest increase in proportion after infection, while the proportions of NK cells and T cells have decreased accordingly. The proportion of macrophages in the uterus did not increase as significantly as that in the lungs. The main immune cells that decreased were T cells (**revised 2nd edition Supplementary Figure 7a, c, e and g**).

Revised Supplementary Figure 7 (a, c, e and g). A single-cell atlas of the maternal immune response to influenza infection in the lung and decidua.

Second, in the revised 2nd edition, we re-analyzed immune cell frequencies in the lungs, peripheral blood, and decidua in the H1N1 post-infection model. Our data showed that Infection induced a robust expansion of CD11b⁺ myeloid cells (macrophages, neutrophils and monocytes) in the lung. By contrary, we did not observe expansion of CD11b⁺ myeloid cells in the decidua (**revised 2nd edition Supplementary Figure 6**), suggesting that IL1R2⁺myeloid cells subset rather than the whole myeloid cells increased in decidua.

Revised Supplementary Figure 6. Differential alterations in myeloid cell populations in maternal lungs, peripheral blood and deciduas following influenza infection.

Third, to specially identify the most relevant macrophage subpopulations in decidua, decidual macrophages were categorized into distinct subsets. An analogous immunoregulatory macrophage subset, exhibiting high *Il1r2* expression, was identified in the decidua. Its abundance increased after infection and decreased with sitagliptin treatment, and its *Il1r2* expression mirrored this trend, closely recapitulating the pulmonary response. This suggests that these cells may serve as a critical link between the lung and maternal-fetal interface, and constitute a local source of IL1R2 (revised 2nd edition Figure 5).

Revised Figure 5 (g-l). Inhibition of DPP4 reduces migratory and *Il1r2*-high macrophages following influenza virus infection.

3) Do you see the same impacts on the uterus if myeloid cell migration is blocked post lung infection (e.g. use of CCR2 KO mice, i.v. clodronate depletion of monocytes)? This could really help clarify whether it is cellular trafficking that is essential.

Reply: Thank you for this kind suggestion. According to your suggestions, we have tried to use clodronate liposomes to effectively depleting monocytes and macrophages. However, clodronate liposomes injection to pregnant mice cause severe fetal loss, suggesting this macrophage depletion agent is not suitable for pregnant mice.

Then, we further used anti-CSF1R antibody to depleted macrophages. Specifically, pregnant mice received an intraperitoneal (i.p.) injection of 300 μg αCSF1R at embryonic day 5.5 (E5.5). At E6.5, the mice were intranasally infected with either H1N1 PR8 influenza virus or sterile saline. To maintain macrophage depletion throughout the experimental period, additional i.p. injections of 300 μg αCSF1R were administered once daily at E7.5, E9.5, and E11.5. All mice were euthanized at E12.5 for tissue collection and evaluation of experimental endpoints. Our data indicated that H1N1+anti-CSF1R antibody group showed more severe lung pathology, indicating higher weight and gross morphological lesions. Meanwhile, we found more severe fetal growth restriction in H1N1+anti-CSF1R antibody group (**Response Figure 1 as follows**). This might be because macrophages are crucial for the lungs in clearing viruses, and it has been previously reported that macrophages are also very important for embryo implantation and development. Therefore, the group that had all the macrophages removed showed more severe lung lesions, and the loss of macrophages also affected embryonic development. This might

indicate that eliminating all the macrophage populations is not an appropriate approach. Thank you very much for your outstanding insights.

Response Figure 1. Removing macrophages aggravate lung lesions and restricted embryonic growth.

4) Does the growth restriction of the fetus carry forward to smaller pups when born? If you let the infected mice recover and give birth, would the neonates show defects in growth or health?

Reply: Thank you for this suggestion. We kept on observing the dams until successful delivery following a 5-day therapeutic regimen and their offspring until postnatal day 21. Our findings demonstrate that sitagliptin treatment significantly alleviated infection-induced maternal weight loss. Postpartum monitoring revealed that infected mothers exhibited lower body weight on days 7 and 14 after delivery, while sitagliptin administration mitigated this decline. Notably, influenza virus infection markedly reduced the number of successfully delivered offspring. The

litter size from healthy pregnancies was generally above 6 pups, whereas infection without treatment resulted in fewer than 5 surviving pups. Sitagliptin treatment substantially reversed this impairment, restoring litter sizes to levels comparable to those of uninfected dams. These results further suggest that in untreated infections, some fetuses sustained severe damage during late gestation, preventing their successful birth. Importantly, the surviving pups from infected but untreated mothers did not show significant differences in health status compared to pups from either uninfected or treated mothers, indicating that sitagliptin not only confers protective effects on the dams but also supports successful delivery and early survival of offspring.

Revised Supplementary Figure 3. Inhibition of DPP4 ameliorates the compromised litter size caused by respiratory influenza virus infection.

Collectively, these findings demonstrate that the sitagliptin confers sustained benefits in both maternal recovery and offspring development, supporting its long-term protective efficacy.

Minor:

1) What is the cellular source of the DPP4 production?

Reply: Thank you for raising this important question regarding the cellular sources of DPP4. According to established literature, DPP4 (CD26) is expressed on various cell types relevant to our infection model, including: (1) immune cells, particularly activated T lymphocytes, dendritic cells, and macrophage subsets; and (2) structural cells, such as lung epithelial and endothelial cells, which can upregulate DPP4 in response to inflammatory stimuli^{10,11}.

2) Supplemental Fig 3g purports to measure viral H1N1 protein, but in reality this is mRNA for

a viral protein measured by RT-PCR it seems. The methods and text do not state what protein is measured, and for influenza, viral RNA is a poor indicator of infectious virus. The authors should determine infectious titer of flu in lung with and without the DPP4 inhibitor.

Reply: Thank you for pointing out the virus titers shortage in the 1st edition of our manuscript. To address your concern, we have performed the following key supplementary experiments, the results of which are now included in the revised manuscript (**revised 2nd edition Supplementary Figure 2**): (1) We performed IHC staining for influenza nucleoprotein (NP) to visually confirm the presence of viral protein and its cellular distribution. This analysis demonstrated NP-positive signals exclusively within the infected lungs, with no detection in the decidua. (2) We quantitatively assessed infectious influenza virus titers in lung homogenates from infected mice, with and without DPP4 inhibitor treatment. This assay confirms that infectious virus is present in the lungs and, critically, that sitagliptin treatment does not reduce the pulmonary titer of infectious H1N1. We have corrected the manuscript text and figure legend (formerly Supplementary Fig 3g) to accurately state that the data derived from qPCR measurement of the influenza matrix gene (viral RNA), not viral protein. The Methods section has been updated with detailed protocols for the plaque assays.

Revised Supplementary Figure 2. Influenza viral replication in the lung and decidua following sitagliptin treatment.

These comprehensive virological data—encompassing infectious titer (plaque assay), viral protein localization (IHC), and genomic presence (qPCR)—collectively provide unambiguous evidence that the protective effects of DPP4 inhibition are mediated by modulating the host's immunopathological response, not by direct antiviral activity. We are grateful to you for

prompting these essential clarifications and for significantly strengthening the rigor of our study.

3) What does the red font indicate in Figures 1c and d?

Reply: Thank you for this question. The red font in Figure 1c highlights four cytokines (IL-28, MIP-3 α , IL-6, and IL-21) that were significantly upregulated in lungs following respiratory influenza virus infection. In contrast, these same four cytokines were downregulated in the decidua after infection. This indicates that the lung and decidua exhibit opposite immune states following infection: the lung is in an immune-activated state, while the decidua is in an immune-suppressed state. We have added the description in the **figure legend** as follows: “The red font denotes four cytokines showing opposite expression between lungs and deciduas.”

4) What was the sublethal dose of PR8 used?

Reply: Thank you for raising this important point regarding experimental details. The sublethal dose of influenza A/Puerto Rico/8/34 (PR8) virus used for intranasal infection in pregnant mice was 16 HA in 50 μ L of PBS. This dose was carefully titrated in preliminary studies to consistently induce significant clinical symptoms (e.g., weight loss, lung inflammation) and robust immune activation without causing mortality in pregnant mice, thereby fulfilling the criteria of a "sublethal" challenge. As suggested, we have now explicitly included this information in the revised "Methods" section (**Viral Infections**) as follows: “At E6.5, pregnant mice were anesthetized and intranasally inoculated with a sublethal dose of 16 HA of PR8 in 50 μ L sterile saline”.

5) The mechanistic link between how DPP4 inhibition limits IL1R2 accumulation in the uterus or even the lung is not defined, or speculated upon. The authors admit this in the discussion, but a hypothesis would be interesting to speculate upon.

Reply: We sincerely thank you for this profound and insightful question. We fully agree that elucidating the mechanistic link between DPP4 inhibition and the reduction of IL1R2 represents a crucial direction for future research. While existing data have clearly established this association, the underlying mechanisms require further in-depth exploration. We propose the following two complementary hypotheses, which have been incorporated into the revised discussion section to stimulate broader research interest.

DPP4 (CD26) cleaves various chemokines and cytokines, thereby modulating their biological activity. Notably, the enzymatic action of DPP4 does not always lead to complete inactivation of chemokines; in some cases, truncated forms of chemokines generated by DPP4 cleavage acquire new functional properties^{10,12}. For example, while truncated forms of CXCL10 and CXCL11 lose their ability to mediate T cell chemotaxis via CXCR3, they may instead recruit myeloid populations such as neutrophils or monocytes by interacting with other receptors^{13,14}. We speculate that the upregulation of DPP4 following viral infection may alter the composition and function of local chemokines, promoting the abnormal accumulation of IL1R2⁺ myeloid precursor cells at inflammatory sites such as the lungs. Conversely, DPP4 inhibition may reverse this process by restoring chemokine homeostasis, thereby limiting the

infiltration of immunosuppressive cells. Additionally, since DPP4 is involved in T cell activation¹⁵, its inhibition may help shift the overall immune environment toward an anti-inflammatory state, further influencing tissue-resident macrophages and other myeloid cells to differentiate away from an IL1R2-high immunosuppressive phenotype.

IL1R2 functions as a critical "decoy receptor" that negatively regulates IL-1 β signaling by sequestering the cytokine¹⁶. Multiple studies suggest a potential link between DPP4 activity and IL-1 β levels: exogenous DPP4 has been shown to reduce IL-1 β in rheumatoid arthritis models, whereas long-term DPP4 inhibition in non-obese diabetic mice downregulates circulating IL-1 β concentrations^{17,18}. These effects may stem from DPP4's indirect regulation of immune cell infiltration and the overall inflammatory milieu. We further hypothesize that DPP4 inhibitors, by broadly mitigating tissue inflammation and cytokine release, attenuate the drive for compensatory upregulation of the decoy receptor IL1R2. From this perspective, the DPP4 inhibition-induced reduction in IL1R2 can be viewed as a consequence of systemically quelling the cytokine storm, leading to a corresponding downregulation of endogenous immunosuppressive feedback mechanisms.

In summary, we propose that DPP4 inhibitors do not act on a single target but rather reshape the systemic and local immunopathological landscape, ultimately normalizing the aberrant accumulation of IL1R2-high immunosuppressive cells. We have added discussion as follows **(Page 11-12 and Line 471-479)**: "We propose that DPP4 inhibition may reduce IL1R2 accumulation through two non-exclusive mechanisms. First, by preventing the cleavage of chemokines such as CXCL10 and CXCL11, DPP4 inhibition may preserve their canonical functions and prevent the aberrant recruitment of IL1R2⁺ myeloid precursors to inflammatory sites^{13,14}. Second, DPP4 inhibition broadly attenuates the inflammatory milieu, thereby reducing the compensatory drive for IL1R2 upregulation as a feedback brake¹⁷. Thus, DPP4 inhibition likely acts by resetting the dysregulated immunopathological environment, rather than through a single target, ultimately normalizing this immunosuppressive axis. However, the precise pathways through which DPP4 inhibition modulates *Il1r2* transcription remain to be elucidated."

Reviewer #3 (Remarks to the Author):

The authors show that gestational influenza leads to immune activation in the uterus defined by activation of inflammatory cytokines leading to intrauterine growth restriction in mice. Using transcriptomic analysis the authors show that systemic DPP4 upregulation (lung, decidua and blood) is associated with adverse outcome. Inhibition of DPP4 with Sitagliptin improves adverse outcome as defined by reduced inflammation and reduced growth restriction. This is an important study elucidating key mechanisms involved in lung infection mediated complications during pregnancy. The methodology used is sound and the conclusions drawn mostly supported by the data. However, there are major concerns that dampen my enthusiasm.

Reply: We really appreciate your comments.

Major comments:

#1: DPP4 is a known protein associated with signal transduction, immune regulation and glucose metabolism (including gestational diabetes). In all the animal experiment that were involving Sitagliptin treatments, the authors did not include the control with the treatment alone. Since Sitagliptin has per se an effect on the immune function would have been important to compare the H1N1+ Sitagliptin group vs Sitagliptin, instead of only not infected. This is a major drawback of the study. These experiments are essential to support the major conclusion of the manuscript.

Reply: We apologize for this oversight. According to your suggestions, we have repeated the influenza infection therapeutic experiment with a complete set of four groups: Saline control, Sitagliptin alone, H1N1 infection alone, and H1N1 infection + Sitagliptin treatment. The results demonstrate that compared to the Saline control, sitagliptin treatment alone in pregnant mice induced no significant alterations in maternal weight gain, lung weight, or pulmonary pathology. Furthermore, assessment of the fetuses and the maternal-fetal interface confirmed that sitagliptin alone did not cause fetal growth restriction, impaired angiogenesis, or disrupted trophoblast invasion.

Revised Figure 2. Inhibition of DPP4 alleviates intrauterine growth restriction caused by respiratory influenza virus infection.

These new data establish two critical points: First, the DPP4 inhibitor sitagliptin exhibits a favorable safety profile for both the dam and fetal development under physiological conditions. Second, the baseline immunomodulatory effect of sitagliptin during normal pregnancy is minimal, suggesting its primary role in our model is to “reset” the infection-triggered aberrant immune state rather than to profoundly perturb the normal immune environment of pregnancy.

We have incorporated these complete experimental results into **Figure 2** in the revised manuscript. We are profoundly grateful to you for this insightful suggestion, which has significantly enhanced the rigor and persuasiveness of our study.

#2: The authors state (lines 240-243): “*Il1r2* mRNA was strongly induced in the lung but not in the uterus (Fig. 3c, f), these results suggest that respiratory virus infection induces IL1R2 expression in lung-resident myeloid cells, which then disseminate through the bloodstream and accumulate systemically, including at the maternal-fetal interface.”

In order to evaluate and distinguish the different subsets of myeloid cells, it would be important to include some key markers that better characterize resident and infiltrating cells. The macrophages subsets in the decidua are characterized not only by the expression of CD11b and F4/80 (classical macrophages markers), but also by CD11c, CX3CR1, TREM2 and a low expression of Ly6C, while infiltrating macrophages travelling via the bloodstream express CCR2.

Reply: Thank you for this important suggestion. In order to evaluate and distinguish the different subsets of myeloid cells, we using single-cell RNA sequencing of CD45⁺ cells in the lung and uterus from dams at E12.5 following treatment with saline, H1N1 infection alone, or H1N1 infection plus sitagliptin. We identified that macrophages constituted the most abundant subset and were significantly expanded following H1N1 infection, an effect that was substantially reversed by sitagliptin treatment. Further analysis revealed a marked H1N1-induced upregulation of *Il1r2* in lung macrophages, which was also significantly suppressed by sitagliptin. In the decidua, we identified ten immune cell clusters. Analysis of the major macrophage population showed a congruent pattern: H1N1 infection robustly induced *Il1r2* expression, and this induction was attenuated by sitagliptin. These findings underscore macrophages as a key *Il1r2*-expressing population during influenza infection in pregnancy.

Revised Supplementary Figure 7. A single-cell atlas of the maternal immune response to influenza infection in the lung and decidua.

We next re-clustered macrophages and searched for the most relevant macrophage subpopulations. A subpopulation of immunoregulatory macrophages highly expressed *Il1r2* was rare under homeostatic conditions but expanded significantly after H1N1 infection, with its proportion subsequently normalized by sitagliptin. Visualization of *Il1r2* expression confirmed its specific enrichment in this immunoregulatory subset after infection, which was abrogated by treatment, suggesting these cells may serve as a key target of sitagliptin and represent a primary source of IL1R2. An analogous immunoregulatory macrophage subset, also exhibiting high *Il1r2* expression, was identified in the decidua. Its abundance increased after

infection and decreased with sitagliptin treatment, and its *Il1r2* expression mirrored this trend, closely recapitulating the pulmonary response. This suggests that these cells may serve as a critical link between the lung and maternal-fetal interface, and constitute a local source of IL1R2 (revised 2nd edition Figure 5).

Revised Figure 5. Inhibition of DPP4 reduces migratory and *Il1r2*-high macrophages following influenza virus infection.

According to your suggestions, we further analyzed distinct molecular signature among these macrophages: beyond *Il1r2*, this macrophage subpopulation specifically expressed notably key mediators of myeloid cell migration and chemotaxis (*Retnlg*, *Cxcr2*, *Mmp9*) and other immunomodulatory genes (*Chil1*, *Steap4*, *Nlrp12*), implying enhanced migratory potential and immunosuppressive ability¹⁹⁻²¹ (revised 2nd edition Figure 5). We also observed that most macrophages in lung high expressing CCR2 while low expressing Ly6C, indicating these cells are monocyte-derived macrophages (Response Figure 2). These scRNA data together identifies tissue-homing IL1R2⁺ macrophages that link pulmonary infection to maternal-fetal interface.

Revised Figure 5 (m and n). Inhibition of DPP4 reduces migratory and *Il1r2*-high macrophages following influenza virus infection.

Response Figure 2. The phenotype of pulmonary macrophages.

#3: Line 296, the authors state: “These findings parallel the immunological alterations observed in influenza infection and support the notion that respiratory viral infections impair placental function and fetal development via indirect, non-viral mechanisms.”

Although MHV can be used as surrogate model for SARS-CoV-2 pathogenesis, it is not a respiratory borne virus, spreading mainly via fecal-oral way or blood transmission. I would suggest to completely rephrase the section on MHV, including the title, since MHV can not be compared to all respiratory viral infection, although most clinical symptoms are similar. The same counts for the abstract. Most respiratory viruses, including SARS-CoV, have a marked tropism for ciliated cells and they are mainly localized in the respiratory tract, while the infection with MHV is multi-organ. Thus, the statements made on respiratory viruses by including MHV data need to be revised throughout the manuscript accordingly.

Reply: Thank you for pointing out the difference in the natural transmission route of MHV compared to typical respiratory viruses. We have rephrased the relevant sections accordingly.

When introducing the MHV model, we now explicitly state its natural transmission routes (fecal-oral/blood-borne) and how they differ from primary respiratory viruses, while explaining why the intranasally inoculated MHV model remains useful for studying the distal effects of lung inflammation. And we have modified the title and relevant text throughout the manuscript. We have refined the broad term “respiratory coronavirus infection” to more precisely refer to the pulmonary coronavirus infection modeled by intranasal MHV inoculation and its systemic consequences. We avoid directly equating MHV with all naturally respiratory-transmitted viruses, instead emphasizing its utility in modeling pulmonary immunopathology and the subsequent indirect impact on pregnancy. We have correspondingly adjusted the wording in the abstract and conclusion to ensure that inferences based on MHV data are rigorous and accurate.

Minor comments:

#1: In Fig. 2d the significance regarding the H1N1 vs H1N1-Sigtaglipin group is missing (as mentioned in the text line 134-135). Please amend.

Reply: Sorry for this problem. It has been well taken in the revised 2nd edition of manuscript.

#2: A minor comment regarding the gating strategy in Suppl. Fig. 4 A: the gate on CD45+ cells is set with living cells. There was a pre-gating strategy not shown with a marker for living cells (e.g. live/dead) otherwise looks like a gate towards a physical parameter. Please clarify.

Reply: We apologize for insufficiently explaining this point in the previous version. In the revised manuscript, we showed in the gating strategy (**revised 2nd edition Supplementary Figure 6a**).

Revised Supplementary Figure 6a. Differential alterations in myeloid cell populations in maternal lungs, peripheral blood and deciduas following influenza infection.

#3: The authors use the mouse-adapted influenza A/PR/8/34 (H1N1). Would have been interesting to see the effect of the treatment on a more clinically relevant seasonal strain. Please discuss.

Reply: Thank you for this insightful comment regarding the use of seasonal influenza strains. We agree that this is a critical point for assessing the broader relevance of our findings. While our study utilized the mouse-adapted A/PR/8/34 (H1N1) strain for its well-established and reproducible pathogenesis in mice, substantial clinical and experimental evidence supports that the core immunopathological features driving fetal growth restriction are highly conserved across different influenza strains, including seasonal H1N1 viruses. This conservation underpins our hypothesis that DPP4 inhibition represents a universally effective strategy against virus-induced pregnancy complications.

Clinical cohort studies have consistently shown that pregnant women infected with seasonal influenza viruses, including H1N1, are at significantly higher risk of developing severe respiratory complications, such as pneumonia and acute respiratory distress syndrome (ARDS), compared to the general population^{22,23}. These conditions are characterized by intense pulmonary inflammation, which is a direct result of the host's immune response to the virus. Furthermore, infection with these seasonal strains is strongly associated with adverse pregnancy outcomes, including spontaneous abortion, preterm birth, and fetal growth restriction²⁴. The shared clinical severity and poor obstetric outcomes between the PR8 model and human infections with seasonal viruses suggest an overlap in the underlying

immunopathological pathways.

The key mechanism we identified that severe lung inflammation leading to a systemic immunopathology that impairs placental function is not strain-specific. Research indicates that respiratory viral infections, including seasonal influenza, trigger a robust innate immune response characterized by a significant influx of myeloid cells into the lungs²⁵⁻²⁷. This inflammatory cascade and the associated cytokine spillover are central to the distal effect on the placenta. Our data demonstrate that DPP4 inhibition further counteracts the accumulation of immunosuppressive molecule IL1R2 at the maternal-fetal interface by attenuating the overall inflammatory milieu and weakening the immune negative feedback regulation. This mechanism targets the host's shared response to viral threat rather than the virus itself.

Therefore, the efficacy of DPP4 inhibition in our model is contingent upon the host's stereotyped inflammatory reaction to a respiratory viral challenge—a reaction that is consistently observed in severe cases of seasonal influenza infection in pregnant women. Since the therapeutic target is the common immunopathological pathway (lung inflammation → systemic immune dysregulation → placental dysfunction) and not a virus-specific component, we have a compelling rationale to hypothesize that DPP4 inhibition would be equally effective in mitigating pregnancy complications caused by a wide range of influenza viruses, including seasonal strains. We have added discussion as follows (**Page 113 and Line 519-521**): “Second, while the efficacy of the inhibitor has been demonstrated using the mouse-adapted influenza A/PR/8/34 (H1N1) strain, future studies should evaluate its therapeutic effect against more clinically relevant seasonal strains”.

#4: In order to compare the percentages during steady state control and infection the absolute number of cells should have been recorded. During infection, normally, a high amount of cells are recruited and activated and there is an increase in the total amount of CD45+ cells that influence the relative percentage number. Both representative gating strategy for saline and H1N1 condition should be shown.

Reply: Thank you for bring this important point up. In the revised 2nd edition, we re-analyzed immune cell frequencies in the lungs, peripheral blood, and decidua in the H1N1 post-infection model. Our data showed that Infection induced a robust expansion of CD11b⁺ myeloid cells (macrophages, neutrophils and monocytes) in the lung. These data confirmed that a high amount of cells are recruited into the lung. By contrary, we did not observe expansion of CD11b⁺ myeloid cells in the decidua (**revised 2nd edition Supplementary Figure 6**), suggesting that IL1R2⁺myeloid cells subset rather than the whole myeloid cells increased in decidua.

Revised Supplementary Figure 6. Differential alterations in myeloid cell populations in maternal lungs, peripheral blood and deciduas following influenza infection.

#5: In Il1r2-KO model would have been interesting to see how and if the immune cell

composition change in relation to the migration from the lung to the placenta. Please discuss.

Reply: This is an insightful point. We fully agree that investigating potential changes in immune cell trafficking from the lung to the placenta is important.

- 1) According to your suggestions, we have tried to use clodronate liposomes to effectively depleting monocytes and macrophages. However, clodronate liposomes injection to pregnant mice cause severe fetal loss, suggesting this macrophage depletion agent is not suitable for pregnant mice.
- 2) Then, we further used anti-CSF1R antibody to depleted macrophages. Specifically, pregnant mice received an intraperitoneal (i.p.) injection of 300 μg αCSF1R at embryonic day 5.5 (E5.5). At E6.5, the mice were intranasally infected with either H1N1 PR8 influenza virus or sterile saline. Our data indicated that H1N1+anti-CSF1R antibody group showed more severe lung pathology, indicating higher weight and gross morphological lesions. Meanwhile, we found more severe fetal growth restriction in H1N1+anti-CSF1R antibody group (**Response Figure 1 as follows**). This might be because macrophages are crucial for the lungs in clearing viruses, and it has been previously reported that macrophages are also very important for embryo implantation and development. Therefore, the group that had all the macrophages removed showed more severe lung lesions, and the loss of macrophages also affected embryonic development. This might indicate that eliminating all the macrophage populations is not an appropriate approach.

3) In *Il1r2*-KO model, by contrast, both *Il1r2*-WT and *Il1r2*-KO mice exhibited severe lung pathology post-infection, characterized by extensive hyperemia and pulmonary enlargement (**revised 2nd edition Figure 6b-e**). Immunohistochemical staining for influenza nucleoprotein and quantitative analysis of viral gene expression in lung and decidual tissues showed that viral replication and propagation were not significantly affected by *Il1r2* deletion (**revised 2nd edition Supplementary Figure 8**). These results indicated that the macrophages in *Il1r2* KO mice have not undergone significant changes, and this model does not cause an increase in lung inflammation as compared to the anti-CSF1R model. The *Il1r2*-KO model may not affect cell migration, but it might have changed the phenomenon of restricted uterine embryonic growth due to the absence of the IL1R2 protein. Thank you very much for your outstanding insights.

Revised Supplementary Figure 8. Influenza viral replication in the lung and decidua following knocking out *Il1r2*.

#6: the authors should address why they have not used the allogenic mouse model that was shown by various groups to mimic best infection-immunity aspects during human pregnancy (Engels et al., Cell Host and Microbe 2017). Please discuss and cite accordingly.

Reply: Thank you for raising this important point regarding the use of an allogeneic pregnancy model. We appreciate the opportunity to discuss our model choice in light of the seminal work by Engels et al., which elegantly demonstrates that allogeneically pregnant mice better recapitulate the severe influenza pathogenesis observed in pregnant humans⁷.

In our study, we initially employed a syngeneic mating model to first establish and dissect the core immunopathological pathway by which a distal respiratory infection impairs placental development and fetal growth. This reductionist approach allowed us to clearly delineate the causal sequence of lung inflammation, systemic immune dysregulation, placental dysfunction and fetal growth restriction, without the additional complexity of strong anti-paternal immune responses present in an allogeneic setting.

We fully acknowledge, in line with Engels et al., that the allogeneic model—where the maternal immune system adapts to tolerate a semi-allogeneic fetus—more closely mirrors the immunological context of human pregnancy. The cited study clearly shows that allogeneically pregnant mice exhibit more severe influenza infection, characterized by a restricted anti-viral immune response (e.g., impaired CD8⁺ T cell migration to the lung and reduced type I interferon response) compared to syngeneically pregnant or non-pregnant mice. This underscores its superior translational value for studying pregnancy-specific vulnerabilities.

Therefore, in direct response to this valuable comment, we have now included a discussion as follows (**Page 12-13 and Line 516-519**): “We fully acknowledge, as demonstrated by Engels et al., that allogeneic pregnancy models, which incorporate immune tolerance mechanisms toward the semi-allogeneic fetus and better recapitulate the immunological landscape of human pregnancy and hold significant translational value⁷.” We are convinced that our findings in the syngeneic model provide a robust mechanistic framework, and we thank the reviewer for guiding us to strengthen the translational perspective of our work by incorporating this critical discussion.

#7: in the methods section, the humane endpoint used is not mentioned. Please amend.

Reply: Thank you for this important suggestion. We have now amended the “**Methods**” section to include both the humane endpoint criteria and the specific euthanasia method. The added text states: “The established humane endpoints included: (1) weight loss exceeding 25% of initial body weight; (2) severe lethargy, hunched posture, or difficulty moving; (3) inability to access food or water independently; and (4) signs of significant distress such as dyspnea. Upon observation of any of these criteria, mice were immediately euthanized by CO₂ inhalation using a gradual fill method, followed by cervical dislocation for confirmation. All animal experiments were performed according to the guidelines of animal protection law and the approved protocols by the Ethics Committee of the University of Science and Technology of China

(approval number: USTCACUC27120122083) and adhered to the National Animal Research Guidelines (China).” We apologize for this oversight and appreciate your guidance in enhancing the rigor of our manuscript.

Reviewer #4 (Remarks to the Author):

Reply: We really appreciate your comments and contribution.

References in this response:

- 1 Ander, S. E., Diamond, M. S. & Coyne, C. B. Immune responses at the maternal-fetal interface. *Sci Immunol* **4**, doi:10.1126/sciimmunol.aat6114 (2019).
- 2 Creisher, P. S. *et al.* Adverse outcomes in SARS-CoV-2-infected pregnant mice are gestational age-dependent and resolve with antiviral treatment. *J Clin Invest* **133**, doi:10.1172/jci170687 (2023).
- 3 Mor, G. & Cardenas, I. The immune system in pregnancy: a unique complexity. *Am J Reprod Immunol* **63**, 425-433, doi:10.1111/j.1600-0897.2010.00836.x (2010).
- 4 Gomez-Lopez, N., StLouis, D., Lehr, M. A., Sanchez-Rodriguez, E. N. & Arenas-Hernandez, M. Immune cells in term and preterm labor. *Cell Mol Immunol* **11**, 571-581, doi:10.1038/cmi.2014.46 (2014).
- 5 Raj, V. S. *et al.* Dipeptidyl peptidase 4 is a functional receptor for the emerging human coronavirus-EMC. *Nature* **495**, 251-254, doi:10.1038/nature12005 (2013).
- 6 Taguchi, F. & Hirai-Yuki, A. Mouse Hepatitis Virus Receptor as a Determinant of the Mouse Susceptibility to MHV Infection. *Front Microbiol* **3**, 68, doi:10.3389/fmicb.2012.00068 (2012).
- 7 Engels, G. *et al.* Pregnancy-Related Immune Adaptation Promotes the Emergence of Highly Virulent H1N1 Influenza Virus Strains in Allogeneically Pregnant Mice. *Cell Host Microbe* **21**, 321-333, doi:10.1016/j.chom.2017.02.020 (2017).
- 8 Westhoven, S. *et al.* From zoonotic spillover to endemicity: the broad determinants of human coronavirus tropism. *mBio* **16**, e0243725, doi:10.1128/mbio.02437-25 (2025).
- 9 Nakayama, M. & Kyuwa, S. Basic reproduction numbers of three strains of mouse hepatitis viruses in mice. *Microbiol Immunol* **66**, 166-172, doi:10.1111/1348-0421.12961 (2022).
- 10 Enz, N., Vliegen, G., De Meester, I. & Jungraithmayr, W. CD26/DPP4 - a potential biomarker and target for cancer therapy. *Pharmacol Ther* **198**, 135-159, doi:10.1016/j.pharmthera.2019.02.015 (2019).
- 11 Zhang, T. *et al.* The Roles of Dipeptidyl Peptidase 4 (DPP4) and DPP4 Inhibitors in Different Lung Diseases: New Evidence. *Front Pharmacol* **12**, 731453, doi:10.3389/fphar.2021.731453 (2021).
- 12 Mulvihill, E. E. & Drucker, D. J. Pharmacology, physiology, and mechanisms of action

- of dipeptidyl peptidase-4 inhibitors. *Endocr Rev* **35**, 992-1019, doi:10.1210/er.2014-1035 (2014).
- 13 Proost, P. *et al.* Amino-terminal truncation of CXCR3 agonists impairs receptor signaling and lymphocyte chemotaxis, while preserving antiangiogenic properties. *Blood* **98**, 3554-3561, doi:10.1182/blood.v98.13.3554 (2001).
- 14 Proost, P. *et al.* Proteolytic processing of CXCL11 by CD13/aminopeptidase N impairs CXCR3 and CXCR7 binding and signaling and reduces lymphocyte and endothelial cell migration. *Blood* **110**, 37-44, doi:10.1182/blood-2006-10-049072 (2007).
- 15 Ikushima, H. *et al.* Internalization of CD26 by mannose 6-phosphate/insulin-like growth factor II receptor contributes to T cell activation. *Proc Natl Acad Sci U S A* **97**, 8439-8444, doi:10.1073/pnas.97.15.8439 (2000).
- 16 Sims, J. E. & Smith, D. E. The IL-1 family: regulators of immunity. *Nat Rev Immunol* **10**, 89-102, doi:10.1038/nri2691 (2010).
- 17 Han, C. K. *et al.* DPP4 reduces proinflammatory cytokine production in human rheumatoid arthritis synovial fibroblasts. *J Cell Physiol* **236**, 8060-8069, doi:10.1002/jcp.30494 (2021).
- 18 Kanasaki, K. *et al.* Linagliptin-mediated DPP-4 inhibition ameliorates kidney fibrosis in streptozotocin-induced diabetic mice by inhibiting endothelial-to-mesenchymal transition in a therapeutic regimen. *Diabetes* **63**, 2120-2131, doi:10.2337/db13-1029 (2014).
- 19 Nixon, B. G. *et al.* Tumor-associated macrophages expressing the transcription factor IRF8 promote T cell exhaustion in cancer. *Immunity* **55**, 2044-2058.e2045, doi:10.1016/j.immuni.2022.10.002 (2022).
- 20 Chen, A. *et al.* Chitinase-3-like 1 protein complexes modulate macrophage-mediated immune suppression in glioblastoma. *J Clin Invest* **131**, doi:10.1172/jci147552 (2021).
- 21 Allen, I. C. *et al.* NLRP12 suppresses colon inflammation and tumorigenesis through the negative regulation of noncanonical NF- κ B signaling. *Immunity* **36**, 742-754, doi:10.1016/j.immuni.2012.03.012 (2012).
- 22 Louie, J. K., Acosta, M., Jamieson, D. J. & Honein, M. A. Severe 2009 H1N1 influenza in pregnant and postpartum women in California. *N Engl J Med* **362**, 27-35, doi:10.1056/NEJMoa0910444 (2010).
- 23 Critical illness due to 2009 A/H1N1 influenza in pregnant and postpartum women: population based cohort study. *Bmj* **340**, c1279, doi:10.1136/bmj.c1279 (2010).
- 24 Creisher, P. S. *et al.* Suppression of progesterone by influenza A virus mediates adverse maternal and fetal outcomes in mice. *mBio* **15**, e0306523, doi:10.1128/mbio.03065-23 (2024).
- 25 Almanzar, N. *et al.* Vagal TRPV1(+) sensory neurons protect against influenza virus infection by regulating lung myeloid cell dynamics. *Sci Immunol* **10**, eads6243, doi:10.1126/sciimmunol.ads6243 (2025).
- 26 Corry, J. *et al.* Infiltration of inflammatory macrophages and neutrophils and widespread pyroptosis in lung drive influenza lethality in nonhuman primates. *PLoS Pathog* **18**, e1010395, doi:10.1371/journal.ppat.1010395 (2022).
- 27 Cabeza-Cabrerizo, M. *et al.* Recruitment of dendritic cell progenitors to foci of influenza A virus infection sustains immunity. *Sci Immunol* **6**, eabi9331,

doi:10.1126/sciimmunol.abi9331 (2021).

We sincerely thank the Reviewers and the Editor for their thoughtful and constructive comments. We have carefully revised the manuscript accordingly and performed additional experiments as requested. Below, we provide a point-by-point response to all the reviewers' comments. Reviewer comments are shown in Blue color, and our responses follow each comment.

REVIEWER COMMENTS

Reviewer #3 and Reviewer #4 (Remarks to the Author):

Inhibition of DPP4 alleviates intrauterine growth restriction caused by respiratory influenza virus infection The authors have adequately addressed all major and minor issues. The addition of requested internal controls as well as the analysis of the different cell subsets strengthened the conclusions drawn on the manuscript has significantly improved. However, one minor issue remains. The analysis of the cell composition in the lung after Sitagliptin administration should have included internal controls as well for better comparability and validation of the data. It would be great if the authors could include this analysis as well to further strengthen the effectivity of their drug candidate.

Reply: Thank you for your positive feedback on the revised manuscript and for your valuable suggestions to further improve the quality of our work. Regarding your comment on “including internal controls for the lung cell composition analysis after Sitagliptin administration,” we have performed immunostaining analysis on lung paraffin-embedded sections from the following four groups: uninfected control, inhibitor-only treatment group, infected untreated group, and infected treated group (**Revised Supplementary Figure 2f-i as follows**). The results showed that compared with the uninfected control group, inhibitor treatment alone did not cause significant changes in lung neutrophils and macrophages, indicating that the DPP4 inhibitor itself does not alter the baseline pulmonary immune landscape. Furthermore, H1N1 infection indeed triggered a significant increase in neutrophils and macrophages, which was markedly alleviated by Sitagliptin treatment. These findings reinforce that Sitagliptin exerts its protective role by modulating the pulmonary immune environment. Once again, we sincerely appreciate your thorough and professional review, which has helped strengthen the rigor and completeness of our conclusions.

Revised Supplementary Figure 2f-i. Influenza viral replication in the lung and decidua and pulmonary immune cells infiltration following sitagliptin treatment.